# *Alternaria alternata* effector AaAlta1 targets CmWD40 and participates in regulating disease resistance in *Chrysanthemum morifolium*

**Shuhuan Zhang[1,2], Lina Liu[1,2], Wenjie Li[1,2], Mengru Yin[1,2], Qian Hu[1,2], Sumei Chen[1,2], Fadi Chen[1,2], Ye Liu[1,2]\*, Zhiyong Guan[1,2]\*, Jiafu Jiang[1,2]\***

**1** State Key Laboratory of Crop Genetics and Germplasm Enhancement and Utilization, Key Laboratory of Landscaping, Ministry of Agriculture and Rural Affairs, Key Laboratory of State Forestry and Grassland Administration on Biology of Ornamental Plants in East China, College of Horticulture, Nanjing Agricultural University, Nanjing, P. R. China, **2** Zhongshan Biological Breeding Laboratory, Nanjing, Jiangsu, P. R. China

\* liuye@njau.edu.cn (YL); guanzhy@njau.edu.cn (ZG); jiangjiafu@njau.edu.cn (JJ)

## Abstract

Black spot diseases caused by the necrotrophic fungal pathogen *Alternaria alternata* adversely affect the growth and yield of many plants worldwide. However, the molecular mechanisms underlying the virulence and pathogenicity of *A. alternata* remain largely unknown. In this study, we report the identification of a novel effector Alta1, secreted by *A. alternata*, which not only contributes to its virulence but also triggers the cell death and defense of the host plant. The expression of Alta1 in *Chrysanthemum morifolium* activated jasmonic acid (JA) signaling, which, in turn, enhanced plant resistance to *A. alternata*. Moreover, we found that Alta1 targeted the WD40-repeat protein of chrysanthemum (*Cm*WD40) after entering host cells. Notably, the *CmWD40* gene showed rhythmic basal expression, and the overexpression of *CmWD40* increased the resistance of chrysanthemum leaves against *A. alternata*, whereas its loss of function led to a decrease in this resistance. The results of the comparative transcriptomics and JA content analyses indicated that *CmWD40* is possibly involved in the accumulation and signaling of JA. The transcript levels of the *MYC2* gene were significantly upregulated in lines overexpressing the *CmWD40* gene compared with that in the wild type. Further, the results of the infection assay revealed that *CmWD40* positively modulated Alta1-induced defense response by activating MYC2 transcription. Overall, the results obtained in this study demonstrate that identified effector Alta1, recognized by the circadian rhythm gene *CmWD40*, triggers JA-induced immune response and enhances disease resistance in chrysanthemum plants.

## Author summary

*Alternaria alternata* is a necrotrophic fungus that causes black spot disease, a severe disease of chrysanthemums (*Chrysanthemum morifolium*). The effectors secreted by *A. alternata* are believed to be the central molecules regulating the complex

**Data availability statement:** The datasets generated during the current study were submitted to the NCBI repository, bioproject PRJNA1110193 and PRJNA1110220.

**Funding:** This work was supported by the National Natural Science Foundation of China (32171854 to ZG, https://www.nsfc.gov.cn), Zhongshan Biological Breeding Laboratory (ZSBBL-KY2024-04) and the Fundamental Research Funds for the Central Universities (QTPY2025005). The funders had no role in study design, data collection and analysis, decision to publish, and preparation of the manuscript.

**Competing interests:** The authors declare no conflicts of interest.

chrysanthemum-*A. alternata* interactions. However, there remains a lack of systematic research on *A. alternata* effectors. In this study, we report the screening and identification of a novel *A. alternata* effector, Alta1, which not only contributes to the full virulence of *A. alternata* but, when overexpressed, also enhances the resistance of the host plant to *A. alternata*. AaAlta1 interacts with CmWD40, encoded by a chrysanthemum rhythm gene, in the nucleus. Moreover, using a combination of transcriptomics, genetics, and molecular biology, we show that the *CmWD40* gene-mediated AaAlta1-induced disease resistance against *A. alternata* involves the activation *CmMYC2* to regulate the jasmonic acid signaling pathway. Our research offers a template for investigating the less explored necrotrophic effectors and their functions in the target host.

## Introduction

A wide variety of pathogenic microbes constantly attack plants. During infection, the pathogen secretes multiple apoplastic proteins that modulate host cell signaling to favor its colonization. To resist attacks from pathogens, plants have evolved two layers of immune responses that confer adequate protection against diverse pathogens [1,2]. First, the pattern-recognition receptors present on the plant cell surface recognize conserved pathogen-associated molecular patterns (PAMP) and activate PAMP-triggered immunity (PTI), thereby constituting the first layer of the immune response [3]. PTI causes a series of rapid immune reactions, including a reactive oxygen species (ROS) burst, callose deposition, $Ca^{2+}$ influx, activation of mitogen-activated protein kinase (MAPK) activity, induced expression of defense genes, and changes in hormone levels, which help resist most pathogens [4–6]. Pathogens release effectors against host cells to invade plants and suppress PTI [7,8], thereby hijacking plant cellular processes to promote pathogen growth and spread, and this phenomenon is known as effector-triggered susceptibility. Plants have evolved *resistance* (*R*) genes that recognize effectors directly or indirectly and activate effector-triggered immunity (ETI), which forms the second layer of the immune response. ETI is stronger than PTI and is often accompanied by a series of resistance responses similar to those in PTI [1–3]. Emerging evidence shows that ETI and PTI can work together to boost plant resistance against disease-causing pathogens [9,10].

Chrysanthemum (*C. morifolium*) is a globally crucial horticultural plant with high economic value because of its ornamental and medicinal uses. However, the black spot disease caused by the necrotrophic fungus *Alternaria alternata* leads to severe losses in annual chrysanthemum production worldwide [11]. After *A. alternata* infection, chrysanthemum leaves initially exhibit a few nearly round brown spots, and then, the lesions increase in size and number, leading to leaf wilting or even plant death [12]. The pathogenic mechanism underlying the chrysanthemum black spot disease is not well understood. Pathogen-secreted proteins are believed to have essential functions in pathogenicity as a fungal pathogen weapon [13]. Pathogens typically use these secreted proteins as effectors to defeat plant defenses. These proteins usually activate PTI or ETI when they contact host cells [13–15]. For instance, the *Verticillium dahliae*-secreted effector, Avirulence on Ve1, is recognized by the immune receptor Ve1 localized on the tomato cell surface and activates plant immune responses [16]. Similarly, another effector, the *V. dahliae*-secreted Asp f2-like protein, interacts with plant U-box 25 (PUB25) and PUB26 E3 ligases to prevent MYB6 from degradation to enhance resistance against *Verticillium* wilt in *Arabidopsis* or cotton [17]. However, the fungal effectors involved in black spot disease have not yet been reported in chrysanthemum.

The allergen protein Alta1 (*A. alternata* allergen 1; Pfam: PF16541), derived from *Alternaria* species, is widely distributed and conserved among the fungal classes Dothideomycetes and Sordariomycetes [18] and is involved in spore germination [19]. Much work has been done to identify and characterize the role of Alta1 in respiratory inflammatory diseases [20]. Notably, a few studies have shown that Alta1 plays a significant role in plant pathology in recent years. A previous study has demonstrated the capacity of Alta1 to bind to flavonoids and proposed a plant virulence mechanism based on its structure [21]. Likewise, studies have demonstrated its ability to bind to and inhibit the pathogenesis-related-5 protein [22]. In addition, a few studies have been performed on its homologs in other plant pathogens, including the characterization of the *Verticillium* Alta1 homolog PevD1 [23–27]. However, the role of Alta1 in plant pathogenesis remains largely unknown.

To study the molecular mechanisms underlying the virulence and pathogenicity of *A. alternata* in chrysanthemum plants, we screened the proteins secreted by *A. alternata*. In this study, we report the identification and characterization of the Alta1 motif-containing effector *Aa*Alta1, which plays a crucial role in pathogen virulence. In addition, *Aa*Alta1 can be recognized by the clock gene *CmWD40* and triggers immune responses by activating MYC2 transcription, thereby the accumulation and signaling transduction of jasmonic acid (JA). The results obtained in this study provide new insights into disease resistance mechanisms and breeding in chrysanthemum plants.

## Results

### Identification of Alta1, a protein secreted by *A. alternata*

To identify defense response effectors secreted by the necrotrophic fungal pathogen *A. alternata*, we performed the genome sequencing of *A. alternata* from infected chrysanthemum leaves in our previous work [28] and the bioinformatics analysis of differentially expressed genes (DEGs) revealed the presence of 114 predicted effectors with unknown functions (S1A Fig). We cloned the selected candidate genes separately into the binary potato virus X vector pICH31160, and the pICH31160: EV (empty vector), and pICH31160:INF1-HA (hemagglutinin) plasmid constructs were used as negative and positive controls, respectively. A transient expression of these genes in *Nicotiana benthamiana* leaves using Agrobacterium-mediated transformation revealed that three of these effectors triggered cell death in *N. benthamiana* 5 days after infiltration (S1B Fig). Notably, the CC77DRAFT_1004449 clone induced strong cell death in *N. benthamiana* leaves (Fig 1). Further bioinformatic characterization of the corresponding *Aa*Alta1 protein showed that it has a typical *A. alternata* allergen 1 (Alta1) motif (Pfam: PF16541) and predicted a secretion signal peptide (SP) within its first 17 residues (Fig 1A). Moreover, it exhibited homology to the *Vd*PevD1 protein of *V. dahliae* (S2A Fig). Furthermore, we performed its comparative analysis with homologous proteins in other *A. alternata* strains using multiple sequence alignment (S3 Fig). We found the homology of these proteins to be higher than 86%, suggesting that the *Alta1* gene and those encoding the effector proteins secreted by other *A. alternata* stains are well conserved.

To further confirm that the Alta1 protein could induce cell death in *N. benthamiana*, Alta1 was produced in *Escherichia coli* BL21 using the pET32a vector (pET32a: AaAlta1), which targeted the protein for secretion into the culture medium. After recovery from the Luria–Bertani culture supernatant using Ni-nitrilotriacetic acid resin, the recombinant protein, Alta1, had a size of 17 kD (Fig 1B). The conformation of the purified Alta1 protein was investigated using crystallography/CD spectroscopy to ensure that the purified protein used for infiltration assays matches the predicted effector protein structure on the KCD web server (https://kcd.cinvestav.mx/) (S4 Fig). The purified Alta1 protein was tested for cell death activity by

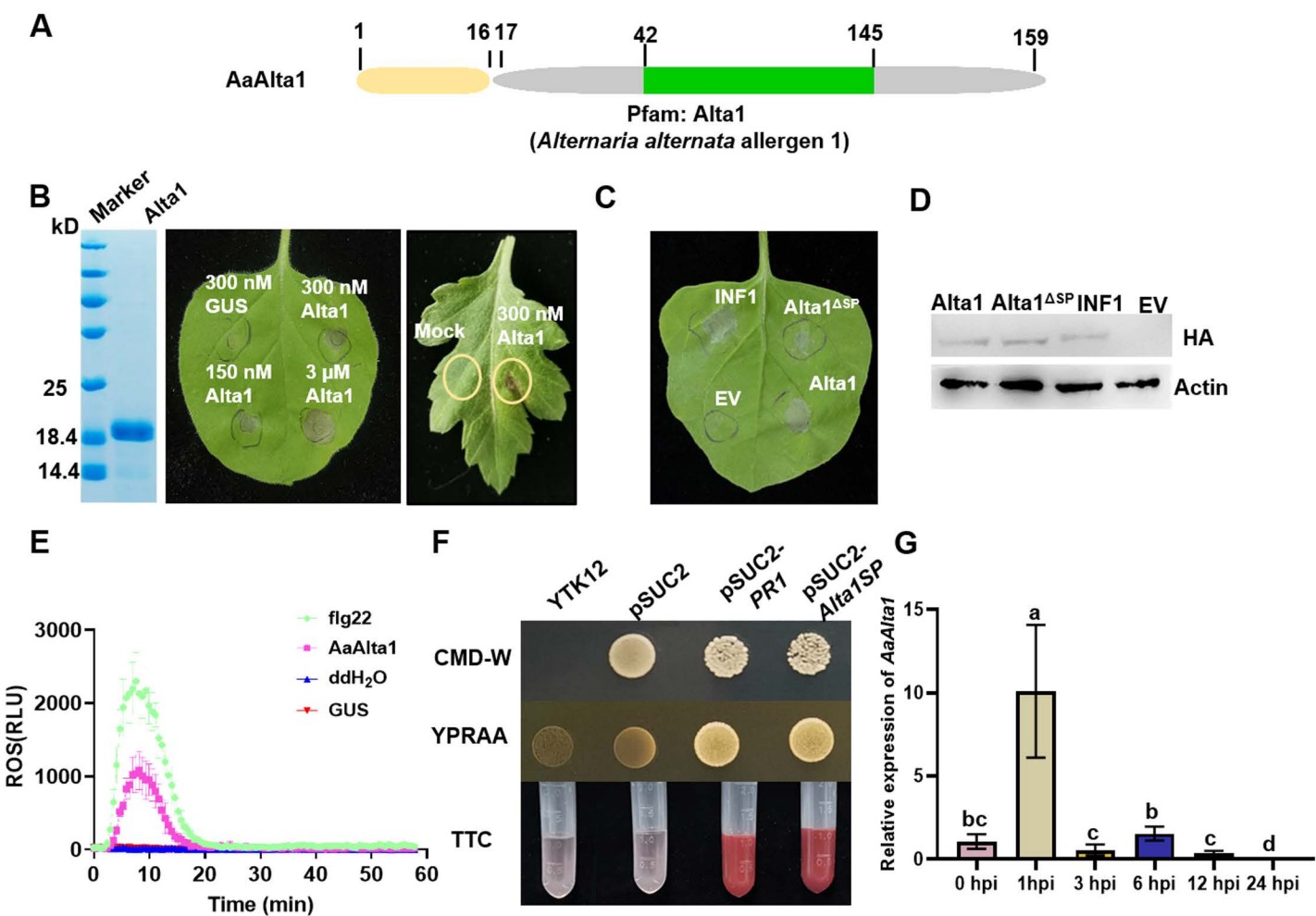

**Fig 1. Identification of *Alternaria alternata* effector *Aa*Alta1. (A)** Schematic diagram of *Aa*Alta1. The 159 amino acid protein has an N-terminal signal peptide (SP) of 16 amino acids for secretion of the mature effector, the *A. alternata* allergen-1 motif sequence. **(B)** AaAlta1 induced cell death in *Nicotian benthamiana* and chrysanthemum. Representative *N. benthamiana* leaves directly infiltrated with purified AaAlta1 protein (300 nM to 3 μM) and photographed 2 days post-infiltration. β-glucuronidase (GUS; control, purified in the same way as *Aa*Alta1) is the culture supernatant control. The image on the right shows chrysanthemum leaves directly infiltrated with purified AaAlta1 and photographed 3 days post-infiltration. **(C)** An SP is required for *Aa*Alta1-induced cell death. *Nicotiana benthamiana* leaves infiltrated with the *Agrobacterium tumefaciens* strain GV3101 cells harboring the genes coding for *A. alternata* effector proteins, *Phytophthora infestans* INF1 gene, and empty vector (EV). The *Agrobacterium* cells harboring the *INF1* gene and EV were used as positive and negative controls, respectively. Images were captured at 7 days. **(D)** Western blot analysis of the transiently expressed proteins obtained from the three agroinfiltrated leaves was performed using an anti-hemagglutinin (HA) antibody. Actin was an equal loading control. **(E)** Luminol-based assay of reactive oxygen species (ROS) burst in the leaves of *N. benthamiana* treated with 1 μM Alta1. Leaves treated with 200 nM flg22 were used as a positive control, whereas those treated with $H_2O$ and 1 μM GUS were used as a negative control. **(F)** Yeast secretion trap assay to verify the functionality of SP in *Aa*Alta1. The yeast *suc2* mutant strain cannot utilize sucrose as a carbon source because of the lack of functional invertase but can use glucose. Colonies were spotted on synthetically defined YPRAA (a medium containing 1% yeast extract, 2% peptone, 2% raffinose, 2 μg/mL antimycin A, and 2% agar) plates with sucrose or glucose and antimycin A. The vector control (EV) and YTK12 were used as a negative control. Pathogenesis-related (PR) 1 represents the SP from the PR1 protein and was used as a positive control. The yeast *suc2* mutant strain expressing pSUC2-Alta1SP was resuspended in 0.1% TTC (2,3,5-triphenyltetrazolium chloride) solution and checked for the formation of red color. **(G)** Relative expression levels of *Aa*Alta1 during infection were assessed by quantitative reverse transcriptase polymerase chain reaction. Data are presented as the mean ± standard error of three biological replicates. Different letters at the top of error bars indicate significant differences ($P < 0.05$, Tukey's test).

infiltrating 300 nM to 3 μM protein solution into the mesophyll of *N. benthamiana* leaves. The results indicate the increasing ability of the Alta1 protein to induce cell death with increasing concentrations of protein solution from 300 nM to 3 μM (Fig 1B). However, treatment with the 300 nM concentration of Alta1 protein solution also induced localized cell death in chrysanthemum. PTI marker genes, such as *PTI5*, *ACRE31*, *WRKY7*, and *WRKY8*, significantly

increased after treatment with Alta1 in *N. benthamiana* (S2B Fig). In addition, the diaminobenzidine (DAB, dark brown) staining and luminol-based assay show that Alta1 triggers ROS burst in *N. benthamiana* (Figs 1E and S2C). Thus, the Alta1 protein can trigger plant cell death and immune responses.

Effectors are secreted through the conventional pathway, requiring the cleavage of the functional N-terminal SP in the endoplasmic reticulum [29]. To investigate the function of *Aa*Alta1 SP, we deleted the N-terminal signal peptide to produce Alta1$^{\Delta SP}$. The ability of expression Alta1$^{\Delta SP}$ to induce cell death was significantly reduced in *N. benthamiana*. In contrast, the expression of Alta1 did trigger cell death (Fig 1C), suggesting that Alta1 may possess specific regions that may help to enter into plant cells. The western blot analysis with anti-HA antibodies showed that Alta1-HA, Alta1$^{\Delta SP}$-HA, and INF1-HA were successfully expressed in plant leaves (Fig 1D). Additionally, we employed a yeast secretion trap assay, fused the sequence encoding the putative Alta1 SP and PR1 (the SP from pathogenesis-related protein 1 protein, as a positive control) with the sucrose invertase sequence in the pSUC2-MSP vector and transformed into yeast strain YTK12 lacking a secreted invertase. As expected, the two transformants and a negative control strain harboring the empty vector (pSUC2) thrived on tryptophan (W) dropout Castenholz medium D (CMD/−W) plates. Nevertheless, only Alta1 SP and PR1 transformants gained the ability to grow on YPRAA (a medium containing 1% yeast extract, 2% peptone, 2% raffinose, 2 μg/mL antimycin A, and 2% agar) plates, which only support the growth of yeast with secreting invertase (Fig 1F). Consistent with this result, the Alta1 SP and PR1 transformants reduced 2, 3, 5-triphenyltetrazolium chloride (TTC) present in the medium into red formazan; however, the untransformed YTK12 strain and the transformant containing empty vector (pSUC2) did not (Fig 1F), confirming that Alta1 SP and PR1 secrete invertase. These results showed that Alta1 SP confers the secretion signal and that Alta1 can be secreted by *A. alternata* into extracellular space and host cells.

Further, we determined the transcript levels of *Aa*Alta1 during *A. alternata* infection. The expression of *Aa*Alta1 was found to be highly induced with 8.4-fold expression at 1 h post-inoculation (hpi) relative to the 0 h time point during the in planta expression analysis in the chrysanthemum cultivar 'Jinba' (Fig 1G). Subsequently, the transcript levels of *Aa*Alta1 decreased. It is speculated that the effector *Aa*Alta1 plays an important role in the infection process of *A. alternata*.

## Effector *Aa*AltA1 contributes to the virulence of *A. alternata*

To determine whether AaAlta1 is involved in the infection or subsequent growth and spread of *A. alternata* in host cells, we constructed two *AaAlta1* gene deletion mutants using the wild-type (WT) *A. alternata* strain, i.e., *Δalta1*, *AaAlta1* knockout (KO) mutants, using polyethylene glycol (PEG)-mediated homologous recombination (S5 Fig). We confirmed the deletion of the *AaAlta1* gene in the mutant strains using the polymerase chain reaction (PCR)-based detection of the *A. alternata* genome. Subsequently, we observed the growth rate and colony morphology of the mutant strains based on radial growth assay. We found that the *Δalta1* mutant strains grew relatively more slowly than the WT strain, but their colonies' morphology resembled that of the WT colonies (Fig 2A). Their hyphae were inoculated onto chrysanthemum leaves. Both *Δalta1* mutant strains produced mild disease symptoms on chrysanthemum leaves compared with those of the WT strain (Fig 2B), reducing the lesion area by approximately 70% (Fig 2C). Notably, when chrysanthemum leaves inoculated with wild-type strains at normal mycelial amounts and *alta1* mutant strains at double mycelial amounts, the results showed that the pathogenicity of the mutant strains with double inoculum was still significant lower than that of the wild-type strains, further demonstrating that the reduction in

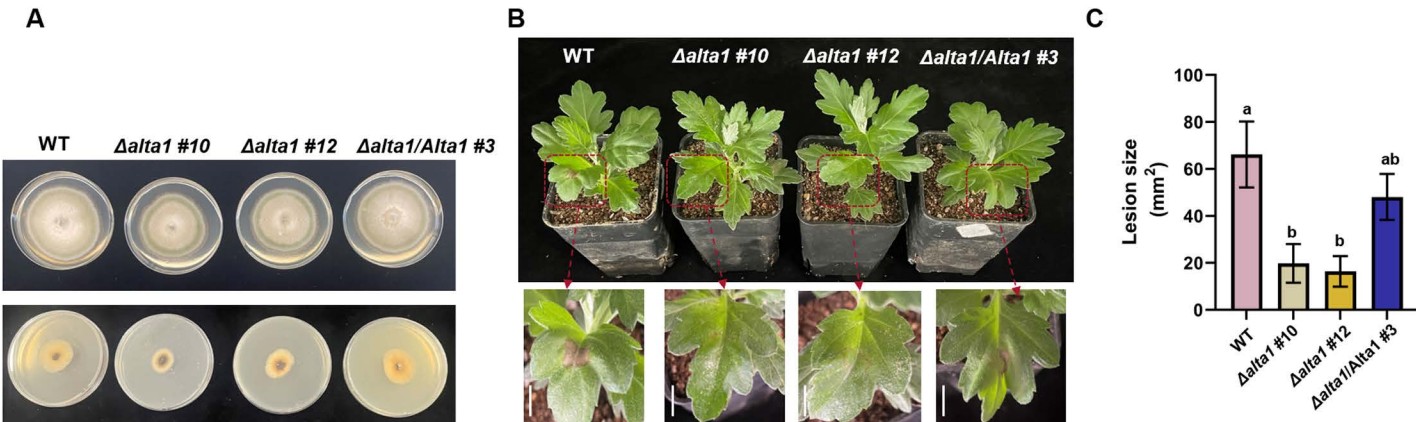

**Fig 2. *Aa*Alta1 is important for the expression of the full virulence of *Alternaria alternata*.** **(A)** Radial growth of *A. alternata* (WT, control), *Δalta1* (*AaAlta1* knockout mutant), and *Δalta1/Alta1* (complementation) strains on potato dextrose agar plates. All indicated fungal strains were grown for 2 and 7 days at 28 °C. The lower panel shows the conidiophore morphology of strains grown for 2 days. The upper panel shows the conidiophore morphology of strains grown for 7 days. **(B)** Symptoms of the black spot disease on chrysanthemum leaves inoculated with WT, *Δalta1* mutant, and complemented strain *Δalta1/Alta1*. Images were captured at 48 h post-inoculation (hpi). Scale bar= 1 cm. **(C)** Disease severity was determined by measuring the lesion area (mm²) of leaves 48 hpi. Data are presented as the mean ± standard error of four biological replicates. Different letters at the top of error bars indicate significant differences (*P* < 0.05, Tukey's test).

lesion size by *alta1* mutant strains was primarily related to the contribution of Alta1 to fungal virulence (S6 Fig). Moreover, to confirm the loss of virulence in the *Δalta1* mutant strain, we generated the *Aa*Alta1 complemented strain (*Δalta1/Alta1*) via PEG-mediated homologous recombination (S5 Fig). The complemented strain rescued the virulence of the *Δalta1* mutant strains and produced relatively more severe disease symptoms on chrysanthemum at a rate similar to that of the WT strain (Fig 2). Additionally, we constructed Alta1 gene deletion mutants in an additional strain of *A. alternata* that causes the tobacco brown spot disease. The inoculation results revealed that the lesion area produced by the two *Δalta1* mutant strains on *N. benthamiana* leaves was significantly reduced compared with that produced by the WT strain (S7 Fig). These results demonstrate that Alta1 is an essential, functionally conserved virulence factor required during the early stages of *A. alternata* infection.

### *Aa*Alta1 triggers the chrysanthemum immune system and defense response

As *Aa*Alta1 can cause rapid necrosis of plant leaves (Fig 1C) and trigger defense responses (S2 Fig), we investigated the *Aa*Alta1-triggered immune responses and disease resistance in chrysanthemum plants inoculated with *A. alternata* after infiltration with 300 nM *Aa*Alta1. The results revealed that *Aa*Alta1-pretreated leaves exhibited enhanced resistance to *A. alternata* compared with that displayed by mock-treated leaves, with mild symptoms and smaller disease lesions than in mock-treated leaves (Fig 3A and 3B).

To investigate the molecular mechanisms underlying Alta1-induced resistance in chrysanthemum plants, RNA-seq analysis was performed on tissues infiltrated with purified *Aa*Alta1 and buffer (Mock). The outcome was a set of 26,884 DEGs in Alta1-treated chrysanthemum leaves, of which 13,769 DEGs were upregulated and 13,115 DEGs were downregulated (Fig 3C). The Kyoto Encyclopedia of Genes and Genomes (KEGG) enrichment analysis showed that the upregulated DEGs were significantly enriched in "MAPK signaling pathway-plant", "alpha-linolenic acid metabolism", and "isoflavonoid biosynthesis" (Fig 3D). Among these are specific genes that are suggested to participate in the plant immune response. The expression levels of such genes, including JA - and stress-related genes, were higher in Alta1- treated than

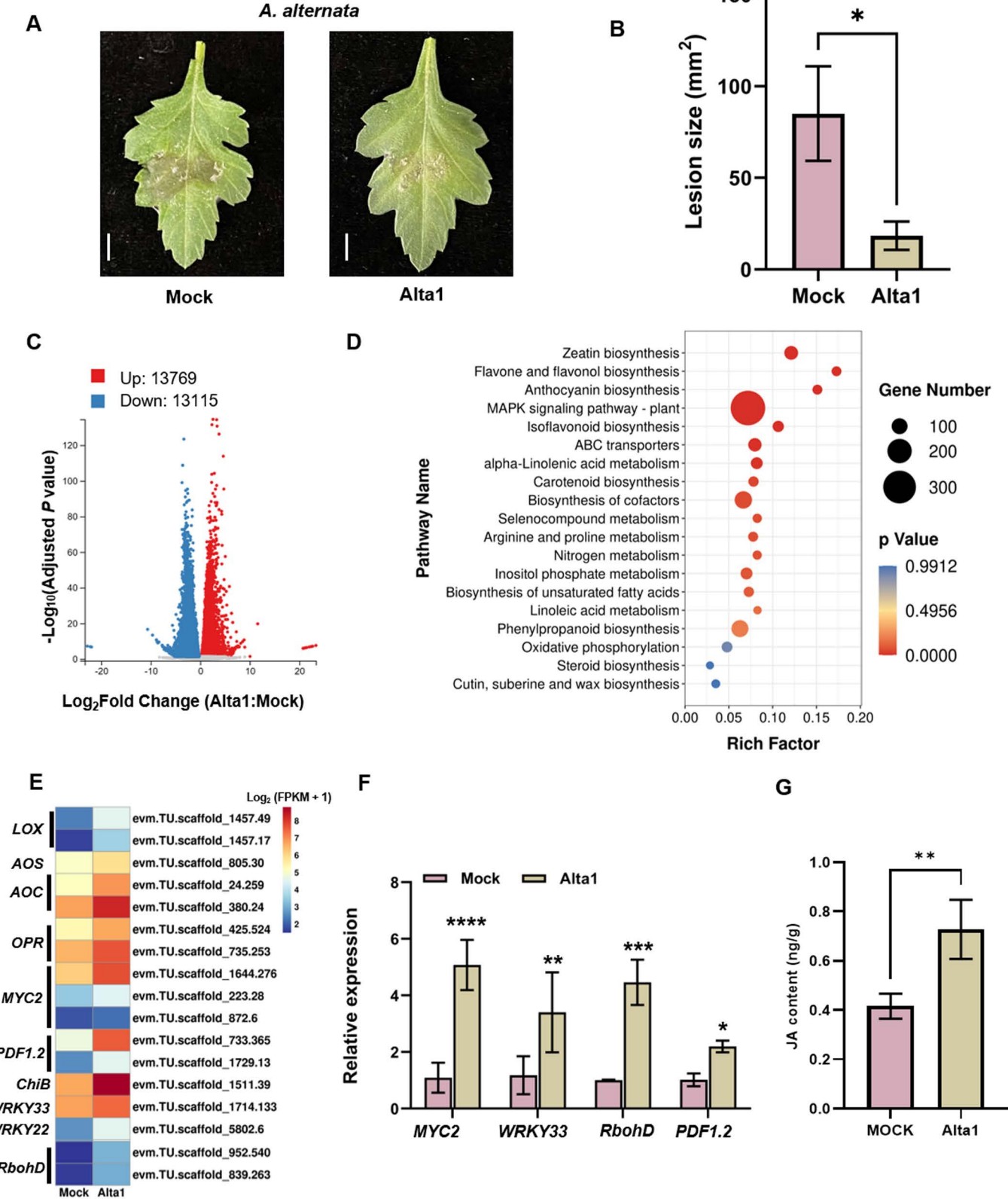

**Fig 3. *Aa*Alta1 induces chrysanthemum resistance to *Alternaria alternata*.** (**A**) The disease symptoms of *A. alternata* in chrysanthemum leaves infiltrated with buffer (Mock) or 300 nM Alta1. Images were captured at 48 hpi. Scale bar=1 cm. (**B**) Mean lesion size on Alta1-infiltrated and mock-infiltrated

chrysanthemum inoculated with *A. alternata* at 48 hpi. Data are presented as the mean ± standard error of four biological replicates. * **P** ≤ 0.05 compared to control, as calculated by one-way analysis of variance (ANOVA). **(C)** Volcano plot showing upregulated (red) and downregulated (blue) genes in the Alta1-infiltrated (Alta1) leaf samples compared with that in the mock-infiltrated (Mock) leaf samples for 24h. **(D)** Kyoto Encyclopedia of Genes and Genomes enrichment analysis of upregulated differentially expressed genes from Alta1- versus mock infiltrated tissues. The top 20 pathways with the most significant P values are shown. **(E)** Heat map of gene expression associated with disease resistance in the transcriptomes. The gene expression values are normalized $\log_2$(FPKM [fragments per kilobase of transcript per million fragments mapped] + 1). From left to right is Mock and Alta1. Mock, mock-infiltrated group; Alta1, 300 nM Alta1-infiltrated group. **(F)** Quantitative reverse transcriptase polymerase chain reaction analysis of *MYC2*, *WRKY33*, *RbohD* and *PDF1.2* genes in mock and Alta1-infiltrated leaves under normal growth conditions. Data are presented as the mean ± standard error of three biological replicates. *P ≤ 0.05 compared with control, as calculated by one-way ANOVA. **(G)** Measurements of jasmonic acid (JA) content in chrysanthemum leaves infiltrated with 300 nM Alta1 or mock solution. Data are presented as the mean ± standard error of four biological replicates. *P ≤ 0.05 compared with control, as calculated by one-way ANOVA.

in the mock-treated leaves (Fig 3E). The quantitative reverse transcriptase-PCR (qRT-PCR) analysis confirmed the expression of *MYC2*, *WRKY33*, *RbohD*, and *PDF1.2* genes (Fig 3F). In addition, JA content was also higher in Alta1-treated leaves than in the mock-treated leaves (Fig 3G). Moreover, JA content was increased with increasing concentrations of Alta1 from 150 nM to 1 mM (S8 Fig). These results suggest that Alta1 triggers the immune system and defense response in chrysanthemum plants by activating JA biosynthesis and signaling.

## *Aa*Alta1 interacts with the chrysanthemum CmWD40

To determine the mechanisms underlying *Aa*Alta1-mediated *A. alternata*–chrysanthemum interactions, we performed a yeast two-hybrid (Y2H) screening of a cDNA library from chrysanthemum leaves infected with *A. alternata* to identify the interactors of *Aa*Alta1. A cDNA fragment showed homology to the *Artemisia annua WD40* transcription factor (ATX64403.1). Subsequently, we cloned the corresponding coding sequence in chrysanthemum, which comprised a 1,095 bp open reading frame sequence, possessing the conserved WD40 repeat motif, and was designated as *CmWD40* (S9 Fig).

To determine whether *Aa*Alta1 directly interacts with *Cm*WD40, we performed a Y2H assay using *Cm*WD40 fused to the activation domain of the yeast transcriptional activator Gal4 and *Aa*Alta1 fused to its DNA binding domain. The results showed *Cm*WD40 interacted with *Aa*Alta1 in yeast (Fig 4A). Additionally, we verified the interactions between *Cm*WD40 and different fragments of *Aa*Alta1, as well as identified Alta1 fragments that induced cell death (S10 Fig). The Y2H assay verified the interactions between *Cm*WD40 and the deletion variants of the *Aa*Alta1 protein containing the *A. alternata* allergen motif, including *Aa*AltA1$^{SP+42–145}$, *Aa*AltA1$^{SP+42–145}$ and *Aa*AltA1$^{SP+42–145}$ (S10A Fig). To define the region required for the cell death-inducing activity of Alta1, we generated a series of N-terminal and C-terminal deletion variants of Alta1 and tested their cell death-inducing activity (S10B Fig). When the allergen motif was present, the deletion variant *Aa*AltA1$^{SP+42–145}$ retained full activity. However, when residues 43–145 were deleted, the cell death-inducing activity was lost. Notably, the homolog of *Cm*WD40 in *N. benthamiana* also interacts with Alta1 in yeast (S11 Fig). Taken together, these data illustrate that the mature effector is only present in the green-highlighted allergen motif, and the allergen motif is indispensable for the interaction with *Cm*WD40 and induction of cell death.

Subsequently, we expressed the glutathione S-transferase (GST)-tagged *Cm*WD40 (GST-*Cm*WD40) and His-tagged Alta1 (His-Alta1) in *E. coli* BL21 and purified them for in vitro pull-down assays. The results of the pull-down assay using GST beads indicated that GST-*Cm*WD40, but not GST, could pull down His-Alta1 (Fig 4B). We further investigated the *Aa*Alta1–*Cm*WD40 interaction *in vivo*. First, we constructed a fungal strain expressing the effector fused to a reporter gene green fluorescent protein (GFP); however, we did not

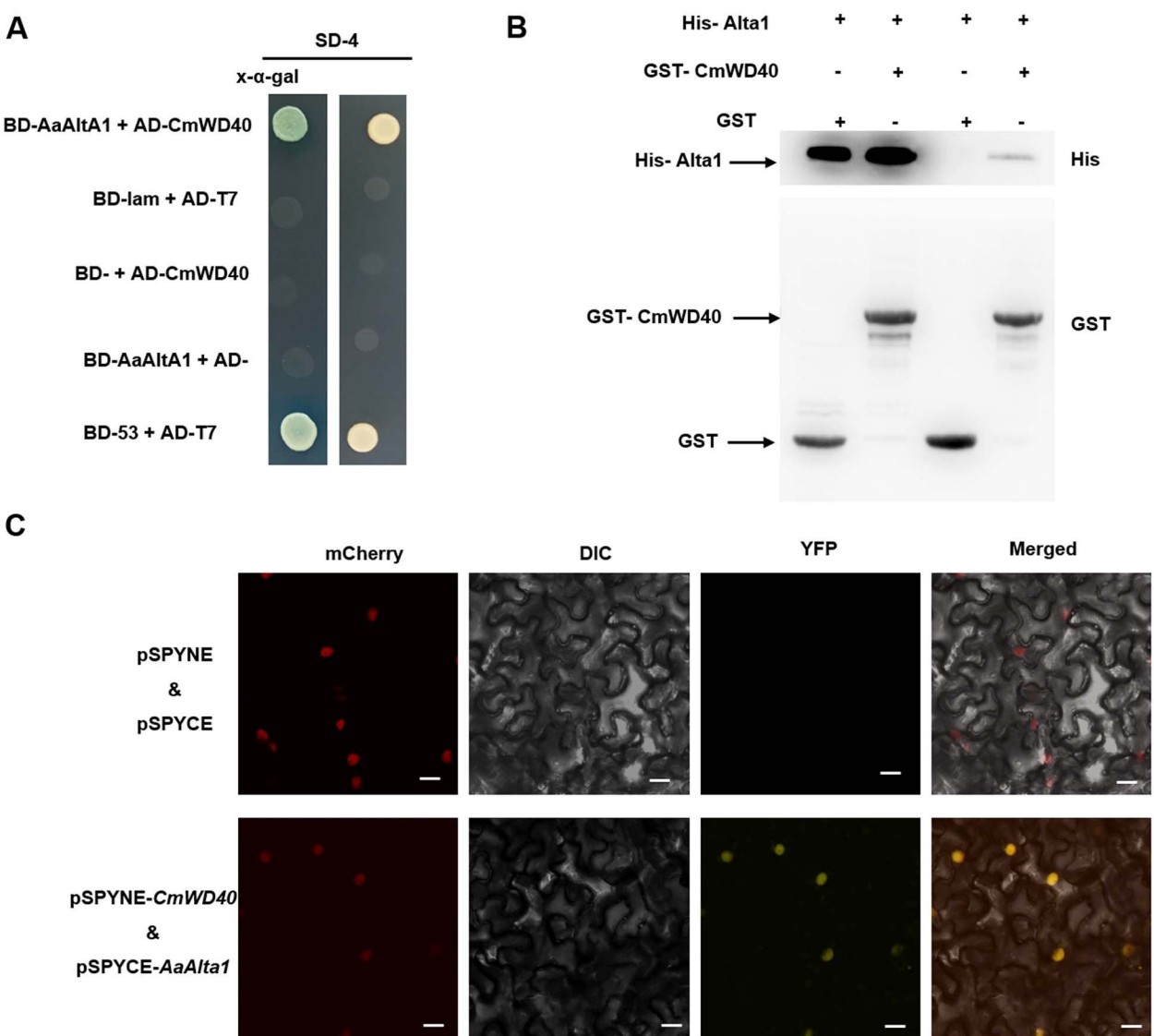

**Fig 4. *Aa*Alta1 physically associates with *Cm*WD40. (A)** Results of the yeast-two-hybrid assay show that *Aa*Alta1 interacts with *Cm*WD40. **(B)** Pull-down assays were performed to test whether *Aa*Alta1 interacts with *Cm*WD40 *in vitro*. The GST-*Cm*WD40 fragment was used to pull down His-Alta1 with the empty GST fragment as a negative control. The GST-CmWD40/His-Alta1 and GST/His-Alta1 complexes were incubated and detected using an anti-His antibody after washes. TheHis-Alta1 could be pulled down using GST-*Cm*WD40 (see the lane designated GST-*Cm*WD40 output) but not using the GST control fragment (see the lane designated GST output). **(C)** A bimolecular fluorescence complementation assay confirmed the interaction between *Aa*Alta1 and *Cm*WD40. YFP: yellow fluorescent protein; mCherry: nuclear localization shown using red fluorescent protein (RFP) activity; DIC: differential interference contrast image; merge: overlay of YFP, RFP, and DIC images.

observe green fluorescence after in vivo inoculation experiments (S12 Fig), likely because the expression level of Alta1 may be extremely low [28], which requires further experimental verification. However, we observed the subcellular localization of AaAlta1$^{\Delta SP}$ and CmWD40. The 35S::GFP-*Aa*Alta1$^{\Delta SP}$ and 35S:: GFP-*Cm*WD40 fusion proteins localized to the nuclei and cytomembranes of *N. benthamiana* cells (S13 Fig). However, the bimolecular fluorescence complementation assay results revealed the interaction of *Cm*WD40 with *Aa*Alta1 in the nucleus (Fig 4C). These results suggest that the effector (*Aa*Alta1$^{\Delta SP}$) directly targets CmWD40 inside the plant cell nucleus.

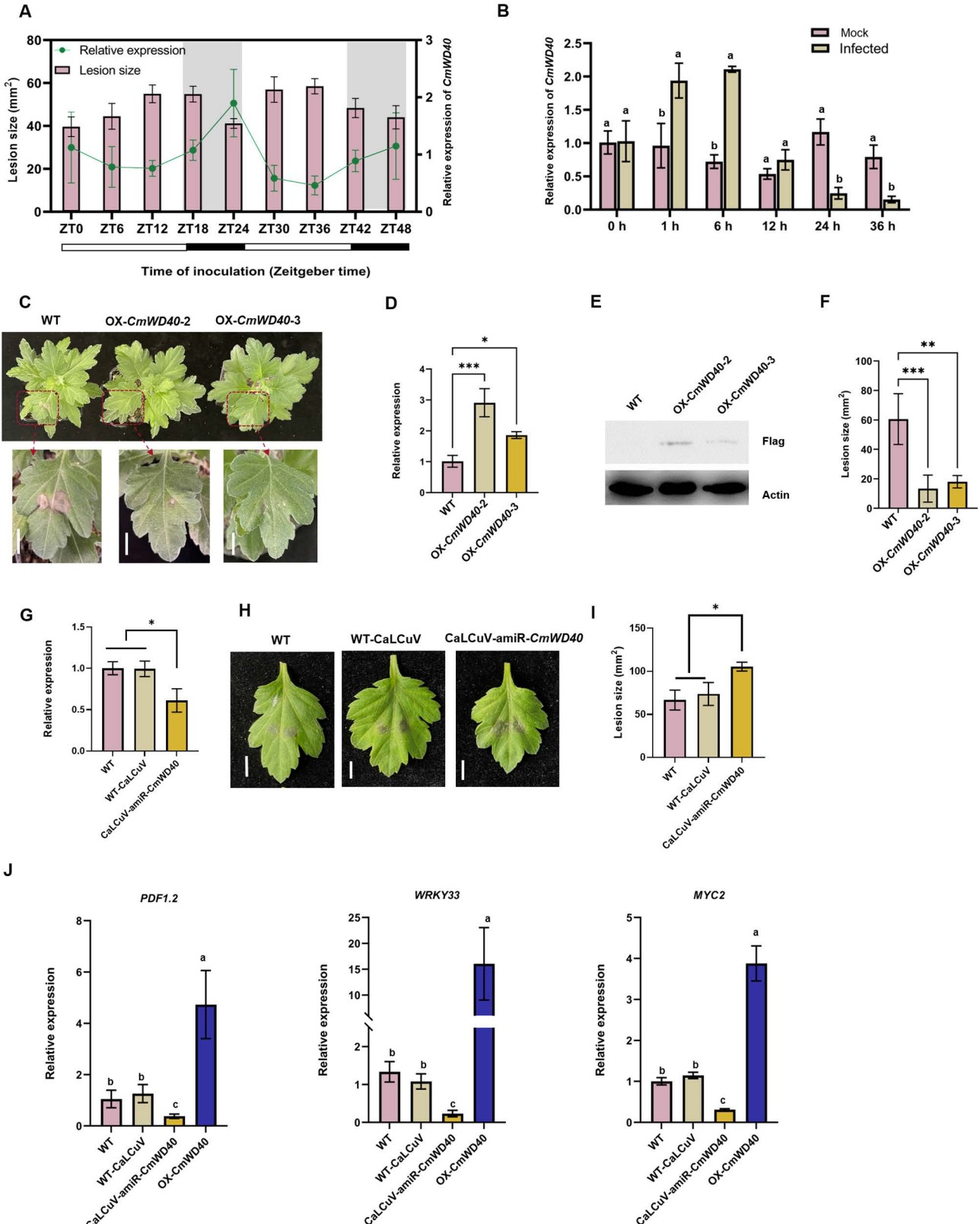

**Fig 5. CmWD40 regulates the resistance of chrysanthemum plants to *Alternaria alternata*.** **(A)** The rhythmic expression pattern of *CmWD40* and the temporal variation in chrysanthemum susceptibility to *A. alternata*. The line chart represents the rhythmic expression pattern of *CmWD40* in

wild-type (WT) plants under long-day (LD) conditions. Leaves were collected every 6 h at the indicated times, and gene expression levels were assessed using quantitative reverse transcriptase polymerase chain reaction (qRT-PCR). The bar chart represents the mean lesion size on chrysanthemum leaves inoculated with *A. alternata* at dawn, i.e., at zeitgeber time (ZT)0 and every 6 h for 48 h under LD conditions. The size of the lesions was statistically analyzed at 48 hpi. Data are presented as the mean ± standard error of three biological replicates. **(B)** Relative expression levels of the *CmWD40* gene during infection were assessed using qRT-PCR. Data are presented as the mean ± standard error of three biological replicates. Different letters at the top of error bars indicate significant differences (P < 0.05, Tukey's test). **(C)** Disease symptoms caused by *A. alternata* infection in chrysanthemum leaves. Leaves of WT and *CmWD40* transgenic lines were inoculated with *A. alternata,* and images were captured at 48 hpi; scale bar=1 cm. **(D)** Identification of the *CmWD40* overexpression (OX) transgenic line (OX-*Cm*WD40) at the transcript level using qRT-PCR. Samples were collected at dawn. Different letters at the top of error bars indicate significant differences (P < 0.05, Tukey's test). **(E)** Immunoblot analysis of *Cm*WD40 in two transgenic lines overexpressing *CmWD40*. Samples were collected at dawn. Total proteins of transgenic seedlings were extracted and detected with anti-Flag antibodies. Actin was used as an equal loading control. **(F)** Mean lesion size on WT and OX-*Cm*WD40 chrysanthemum leaves inoculated with *A. alternata* at 48 hpi. Data are presented as the mean ± standard error of three biological replicates. * $P \leq 0.05$ compared with control, as calculated by one-way analysis of variance (ANOVA). **(G)** Expression of *Cm*WD40 in CaLCuV-amiR-*Cm*WD40 transgenic lines. Samples were collected at dawn. Different letters at the top of error bars indicate significant differences (P < 0.05, Tukey's test).**(H)** Disease symptoms caused by *A. alternata* infection in chrysanthemum leaves. Leaves of the WT and cabbage leaf-curl geminivirus vector (CaLCuV)-amiR-*Cm*WD40 lines were inoculated with *A. alternata,* and images were captured at 48 hpi; scale bar=1 cm. **(I)** Mean lesion size on WT and *CmWD40* silenced chrysanthemum leaves inoculated with *A. alternata* at 48 hpi. Data are presented as the mean ± standard error of three biological replicates. * $P \leq 0.05$ compared with control, as calculated by one-way ANOVA. **(J)** qRT-PCR analysis for expression of *PDF1.2*, *WRKY33*, and *MYC2* genes reveals that their transcript levels are upregulated in OX-*Cm*WD40 and CaLCuV-amiR-*Cm*WD40 transgenic lines, respectively, upon infection with *A. alternata*. Data are presented as the mean ± standard error of three biological replicates. Different letters at the top of error bars indicate significant differences ($P <0.05$, Tukey's test).

## *CmWD40* shows rhythmic basal expression and positively regulates plant resistance against *A. alternata*

The phylogenetic analysis revealed that *CmWD40* is closely related to the light-regulated gene *LWD1* (AT1G12910.1) in *Arabidopsis thaliana* (S9C Fig). As *At*LWD1 is a type of circadian clock component [30,31], we checked whether *CmWD40* shows a rhythmic expression pattern and found that *CmWD40* expressed in a circadian rhythm in the WT 'Jinba' cultivar under long-day (LD) photoperiod (16 h light/8 h dark) conditions, with the rhythmic peak of expression at dawn, i.e., at zeitgeber time (ZT)0, ZT24, and ZT48 (Fig 5A). Subsequently, chrysanthemum leaves inoculated with *A. alternata* were sampled at different times to analyze differences in *CmWD40* transcript levels. The expression of *CmWD40* was highly induced by 1-fold at 1 hpi and 1.2-fold at 6 hpi relative to the 0 h time point (Fig 5B), indicating that its expression was induced by infection with *A. alternata*. Notably, the susceptibility of chrysanthemum plants to *A. alternata* varied with the time of inoculation under LD conditions, with the smallest lesion size observed after inoculation at dawn (Fig 5A), suggesting that the *Cm*WD40 is important for plant disease resistance.

To further confirm the role of *Cm*WD40 in regulating resistance against the chrysanthemum black spot disease, two independent overexpression (OX) chrysanthemum transgenic lines, OX-*Cm*WD40-2 and OX-*Cm*WD40-3, were obtained (Fig 5). The RT-qPCR results revealed that the transcript levels of *Cm*WD40 in the transgenic lines were 2.3 and 1.1 times higher than those in the WT (Fig 5D) and successfully expressed at the protein level (Fig 5E). The results of the inoculation assays showed that the *CmWD40* overexpressing lines, OX-*Cm*WD40-2 and OX-*Cm*WD40-3, exhibited enhanced resistance to black spot disease (Fig 5C), with the lesion area reduced by 78 and 70%, respectively (Fig 5F). In addition, we transiently silenced *CmWD40* in chrysanthemum using a modified cabbage leaf-curl geminivirus vector (CaLCuV) harboring amiR-*Cm*WD40 (CaLCuV-amiR-*Cm*WD40) construct and found that the expression of *CmWD40* in CaLCuV-amiR-*Cm*WD40 infected plants was lower than the relative expression value of 0.59 in the WT and CaLCuV vector-infected plants (Fig 5G). Conversely, the *CmWD40* silenced lines (CaLCuV-amiR-*Cm*WD40) displayed enhanced susceptibility, with an increased lesion area compared with that exhibited by the WT and CaLCuV vector-infected plants (Fig 5H and 5I). Consistent with these results, the expression

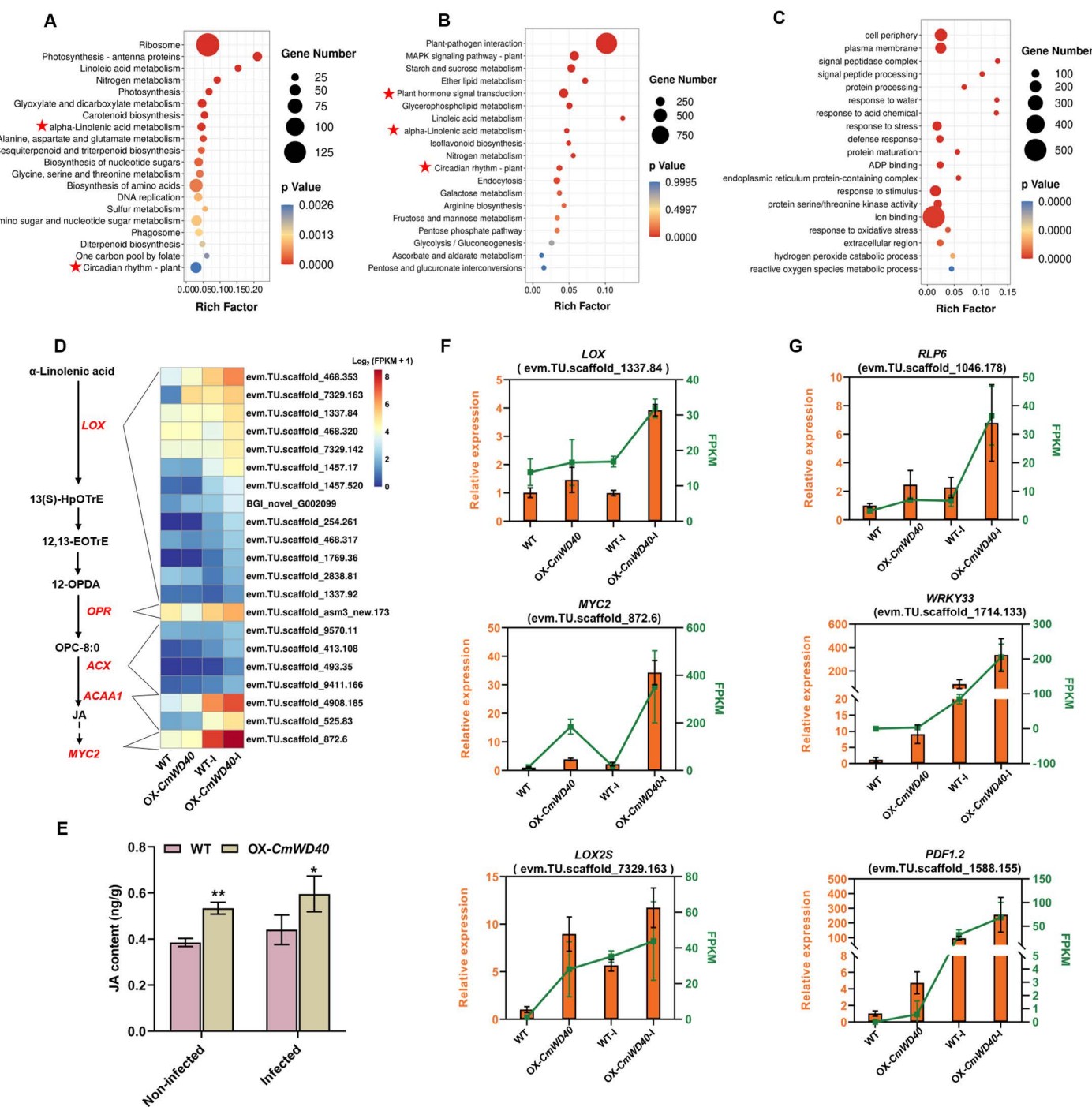

**Fig 6. *Cm*WD40 activates JA biosynthesis and signaling to promote immunity. (A)** Kyoto Encyclopedia of Genes and Genomes (KEGG) analysis of differentially expressed genes (DEGs) in the overexpression (OX)-*Cm*WD40 transgenic line compared with that in the wild type (WT). The top 20 pathways with the most significant *P*-value are shown. **(B)** KEGG analysis of DEGs in the OX-*Cm*WD40 transgenic line compared with that in the WT after *Alternaria alternata* infection for 48 h. The top 20 pathways with the most significant *P*-value are shown. **(C)** Gene Ontology (GO) analysis of DEGs in the OX-*Cm*WD40 transgenic line compared with that in the WT after *A. alternata* infection for 48 h. The top 20 pathways with the most significant *P*-value are shown. **(D)** Heat map of the normalized log₂(FPKM [fragments per kilobase of transcript per million fragments mapped] + 1) values of JA-related genes among the upregulated DEGs. WT, control, non-infected plants (WT 'Jinba'); OX-WD40, non-infected OX-*Cm*WD40 transgenic plants; WT-I, control, infected plants; OX-WD40-I, infected OX-*Cm*WD40 transgenic plants. **(E)** Comparison of JA content in WT and OX-CmWD40 transgenic plants with or without *A. alternata* infection. Data are presented as the mean ± standard error of three biological replicates. * *P* ≤0.05 compared with control, as calculated by one-way ANOVA. **(F)** Expression levels of genes implicated in JA biosynthesis and signal

transduction in different samples. The left vertical axis represents the relative gene expression levels from qRT-PCR (orange), and the right vertical axis represents the FPKM values obtained from RNA-seq (green). **(G)** Expression levels of defense-related genes in different samples. The left vertical axis represents the relative gene expression levels from qRT-PCR (orange), and the right vertical axis represents the FPKM values obtained from RNA-seq (green).

of several PTI- and JA signaling-related genes, such as *PDF1.2*, *WRKY33*, and *MYC2*, was upregulated in OX-*Cm*WD40 and downregulated in amiR-*Cm*WD40 (Fig 5J). These results indicate that *Cm*WD40 positively regulates the resistance of chrysanthemum against necrotrophic fungal pathogen *A. alternata*.

## *CmWD40* positively modulates the JA pathway to promote immunity

To further explore the molecular basis underlying the *Cm*WD40-mediated defense response of chrysanthemum to *A. alternata*, we utilized RNA-seq analysis of the leaves of the *Cm*WD40-2 overexpressing line (OX-*Cm*WD40) and WT chrysanthemum both without and with *A. alternata* inoculation at 48 hpi. DEGs were identified by comparing the OX-*CmWD40* group with the WT group (OX-*Cm*WD40 vs. WT) and the OX-*CmWD40*-I group with the WT-I group (OX-*Cm*WD40-I vs. WT-I). The identified DEGs were classified based on KEGG and Gene Ontology analyses to assess their biological functions. In the OX-*Cm*WD40 vs. WT comparison, the KEGG pathways of "alpha-linolenic acid metabolism" and "circadian rhythm-plant" were significantly enriched (Fig 6A), suggesting that*Cm*WD40 is involved in regulating the expression of circadian rhythm- and JA biosynthesis-related genes under normal developmental process. Notably, the key genes of the circadian clock and JA signaling transduction, such as *CCA1*, *TCP*, and *MYC2*, exhibited the rhythmic expression pattern in the WT and OX-*Cm*WD40 transgenic plants under LD conditions, as assessed using qRT-PCR (S14 Fig). Consistent with these results, *Cm*WD40 participates in the transcriptional activation of some circadian clock-related genes (S15 Fig).

Further, the results of the KEGG analysis showed that after inoculation with *A. alternata*, the pathways of "Plant-pathogen interaction," "MAPK signaling pathway-plant," "plant hormone signal transduction," "alpha-linolenic acid metabolism" and "circadian rhythm-plant" were significantly enriched (Fig 6B), further indicating that JA signaling-, circadian clock- and defense-related genes are regulated by *Cm*WD40. In addition, the Gene Ontology analysis revealed that DEGs were significantly enriched in items related to defense response against biotic and abiotic stresses, including "response to stress," "defense response," "response to stimulus," and "response to oxidative stress" (Fig 6C). During *A. alternata* infection, the OX-*Cm*WD40 transgenic line accumulated much higher levels of the transcripts of defense-related genes, including PTI and ETI-related genes (S16 Fig), supporting an active role of CmWD40 in regulating resistance against black spot disease.

To further verify whether the JA pathway is involved in the elevated resistance of OX-*Cm*WD40 to *A. alternata* infection, the changes in the transcript levels of JA-related genes were analyzed based on RNA-seq. The heat map showed that a few DEGs, including *LOX*, *OPR*, *ACX*, *ACCA1*, and *MYC2*, implicated in JA biosynthesis and signal transduction exhibited higher transcript levels in the OX-*Cm*WD40 line than in the WT both with and without infection *A. alternata* (Fig 6D). Consistently, JA content showed a significant increase in the *CmWD40* overexpression lines, regardless of inoculation with *A. alternata* (Figs 6E and S17). Further experiments showed that two *LOX* genes, involved in JA biosynthesis, and MYC2, a major positive regulator mediating JA signaling, exhibited a drastically increased expression when *CmWD40* was overexpressed and showed a further rising trend after being inoculated with *A. alternata* (Fig 6F). Notably, the expression trends of the downstream defense-related genes, including *PDF1.2*, *RLP6*, and *WRKY33*, were similar to those of the JA signaling-related

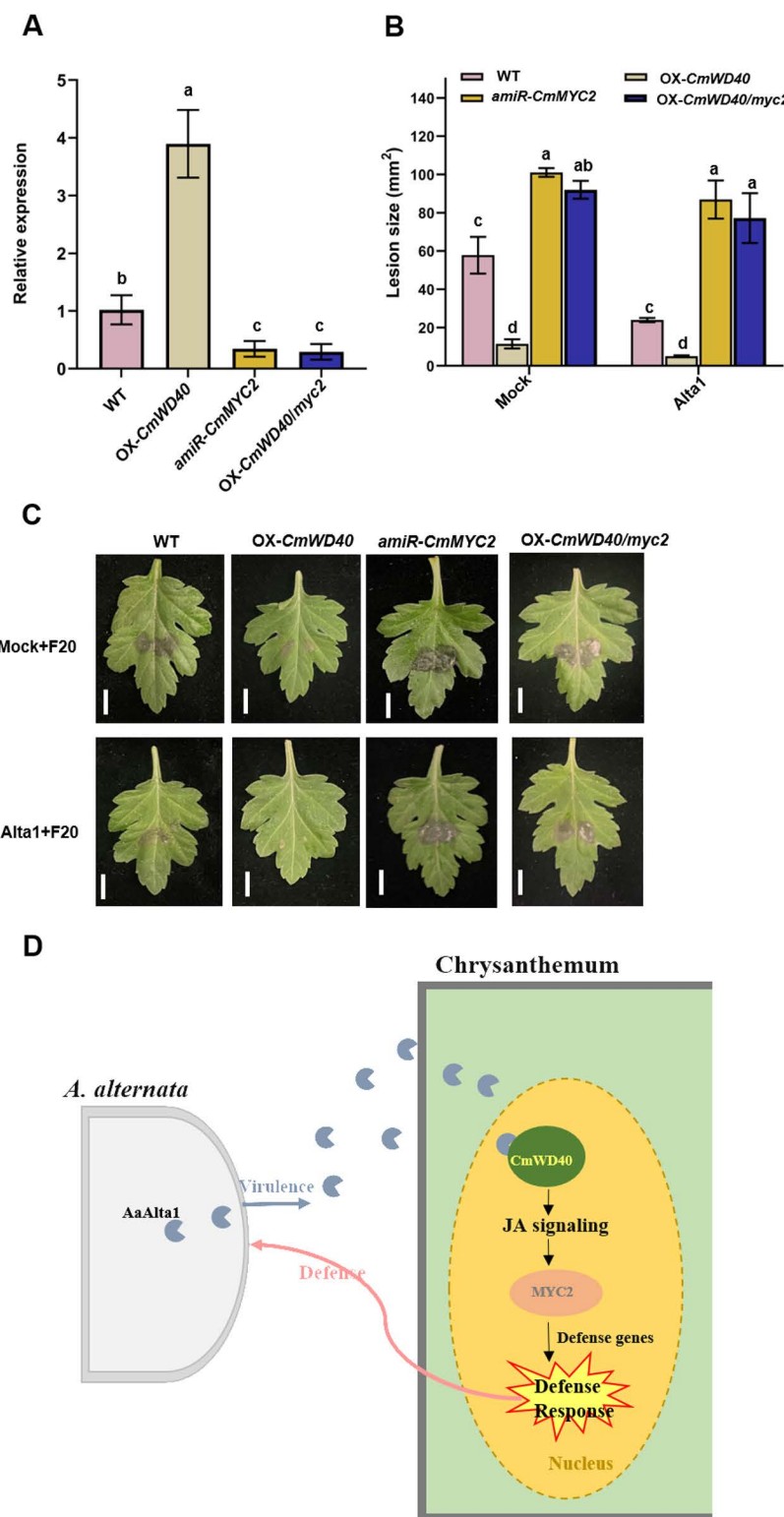

**Fig 7. CmWD40-mediated Alta1-induced disease resistance relies on MYC2 function. (A)** Expression of *Cm*MYC2 in the WT and OX-*Cm*WD40, amiR-*Cm*MYC2, and OX-*Cm*WD40/*myc2* transgenic lines. Data are presented as the mean ± standard error of three biological replicates. Different letters at the top of error bars indicate significant differences ($P$ <0.05, Tukey's test). **(B)** Mean lesion size at 48 hpi on the WT, OX-*Cm*WD40, amiR-*Cm*MYC2, and OX-*Cm*WD40/myc2 lines after pretreatment with mock solution or 300 nM Alta1 and inoculation

with *A. alternata*. Data are presented as the mean ± standard error of four biological replicates. * *P* ≤0.05 compared with control, as calculated by one-way analysis of variance (ANOVA). **(C)** Disease symptoms caused by *Alternaria alternata* infection in chrysanthemum leaves of the WT, overexpression (OX)-*Cm*WD40, amiR-*Cm*MYC2, and OX-*Cm*WD40/*myc2* lines after pretreatment with buffer (mock) or 300 nM Alta1. Images were captured at 48 hpi; scale bar=1 cm. **(D)** Proposed model of *Aa*Alta1 function during *A. alternata*–chrysanthemum interactions. *Aa*Alta1 is recognized by *Cm*WD40, which induces JA signaling to contribute to plant disease resistance.

genes during *A. alternata* infection (Fig 6G). These results suggest that *CmWD40* positively regulates plant resistance by being involved in JA accumulation and signaling transduction.

## CmWD40 mediates AaAlta1-induced defense relies on MYC2 function

The drastically increased expression of MYC2, a master regulator of the JA signaling pathway, in OX-*Cm*WD40 lines (Fig 6F), indicates that Alta1 triggers the immune system and defense response through JA signaling (Fig 3). Moreover, the expression levels of *MYC2* in Alta1-treated plants were significantly higher than those in the mock-treated plants (Fig 3F). These results suggest that the Alta1-*Cm*WD40 module may affect the synthesis of JA by modulating *MYC2* transcript expression, eventually inducing resistance against *A. alternata*. Therefore, we transiently silenced *CmMYC2* in the WT and OX-*Cm*WD40 background using a CaLCuV vector harboring amiR-*Cm*MYC2 construct (amiR-*Cm*MYC2 and *Cm*WD40/ *myc2*) (Fig 7). The expression of *CmMYC2* in amiR-*Cm*MYC2 and *Cm*WD40/*myc2* lines was markedly knocked down, as assessed by qRT-PCR analysis (Fig 7A). Next, the leaves of the WT, OX-*Cm*WD40, amiR-*Cm*MYC2, and *Cm*WD40/*myc2* lines were treated with or without 300 nM *Aa*Alta1 before being inoculated with *A. alternata*, and the sizes of lesions that appeared at 48 hpi were measured. The leaves of the OX-*Cm*WD40 displayed smaller lesion diameters, whereas those of the two *myc2* mutant lines (amiR-*Cm*MYC2 and *Cm*WD40/ *myc2*) displayed higher lesion diameters than those in the WT (Fig 7B and 7C), indicating that OX-WD40-induced defense relies on *Cm*MYC2 function. Additionally, the results revealed that the two *myc2* mutant lines displayed similar susceptibility to *A. alternata* after mock-infiltration or Alta1-infiltration. However, the Alta1-infiltrated OX-*Cm*WD40 lines displayed stronger resistance than that of the mock-infiltrated OX-*Cm*WD40 line (Fig 7B and 7C), suggesting Alta1-induced defense in OX-*Cm*WD40 lines relies on *Cm*MYC2 function. These results suggest that CmWD40 mediated AaAlta1-induced disease resistance against *A. alternata* involves CmMYC2 activation to regulate the JA signaling pathway.

## Discussion

### Role of effector *Aa*Alta1 in plant-pathogen interaction

Plant pathogens use their effectors as a common weapon to disrupt the host immune system in favor of pathogen propagation. As a defense mechanism, plants evolved a surveillance system to detect pathogen effectors and activate innate immunity. Thus, effectors are considered essential molecular markers for identifying the activation or inhibition of plant immunity. However, our knowledge regarding the effectors of the necrotrophic fungal pathogen *A. alternata* remains limited. The allergen protein Alta1 is widely distributed and conserved among the fungal classes Dothideomycetes and Sordariomycetes [18], it plays a role in spore germination [19], even in respiratory inflammatory diseases [20]. Some reports show that Alta1 functions in plant pathology through binding to flavonoids [21], the pathongenesis-related-5 protein [22], etc. In this study, we screened the effectors secreted by *A. alternata* during the infection of chrysanthemum using transcriptomics and bioinformatics analyses (S1 Fig). *Aa*Alta1 was identified as an effector of *A. alternata*-triggered immune responses, and its

interaction with the CmWD40 protein to activate *MYC2* transcription to promote JA signaling is crucial to enhance plant immunity.

Currently, multiple homologous proteins of Alta1 have been reported in other fungal species. For example, an Alta1 homolog in *Magnaporthe oryzae* binds to the plasma membrane of the host plant and triggers necrosis, ROS accumulation, and the expression of multiple defense-related genes, suggesting that it plays the role of a PAMP [32]. Another Alta1-like protein, PevD1 secreted by *V. dahliae,* induces plant cell death and immune responses [23,24] and interacts with *Nb*NRP1 [25,26] or *At*ERF114 proteins [15], inducing disease resistance in plants. A recent study has shown that the effector PevD1 induces leaf senescence by promoting ORE1-mediated ethylene biosynthesis [27]. Here, we found that Alta1 triggers the immune response and increases the content of JA (Figs 1B and S8), the *Δalta1* mutant exhibited markedly reduced virulence on chrysanthemum compared with that displayed by the WT and complemented strain *Δalta1*/Alta1 (Fig 2B and 2C). These results showed that Alta1 family proteins have complicated pathogenic roles in infection.

Recent studies have revealed the ability of a multitude of pathogen effectors to enhance plant disease resistance [33,34]. For example, the overexpression of *V. dahliae* effector VP2 in cotton induces the expression of genes implicated in JA and salicylic acid (SA) signaling and lignin synthesis, enhancing disease resistance [35]. *Trichoderma harzianum* effector Hyd1 can induce maize resistance against *Curvularia lunata* [36]. Likewise, treatment with the *Aa*Alta1 protein enhanced chrysanthemum resistance by activating the JA signaling pathway, phenylpropanoid biosynthesis, isoflavonoid biosynthesis, and ROS pathway (Fig 3), which induced an immune response similar to that caused by *A. alternata* infection (S18 Fig).

The upregulated transcript levels of genes implicated in isoflavonoid biosynthesis (Fig 3D), including those encoding the cytochrome P450, pterocarpan synthase, and 2-hydroxyisoflavanone dehydratase enzymes, etc. (S19 Fig). Isoflavones are a group of plant phenolic compounds, that mediate important interactions with plant-associated microbes, including defense against pathogens. In addition, in some cases, isoflavonoids can act as phytoalexins, providing resistance against infections. For instance, a new isoflavonoid, triticein, catalyzed by a cytochrome P450 enzyme, inhibits fungal spore germination and bacterial growth [37]. In soybeans, glyceollin is a key phytoalexin induced by cell wall glucan elicitors, used to resist infection from *Phytophthora sojae* [38]. In *Medicago truncatula*, medicarpin works synergistically with SA to combat *Erysiphe pisi* and *Rhizoctonia solani* [39]. It can be inferred Alta1-induced genes encode enzymes that catalyze various steps in the biosynthetic pathway leading to the formation of isoflavonoids, which are involved in the production of phytoalexins in certain plants, which, in turn, play a role in plant defense against pathogens. On the other hand, JA plays a major role in the establishment of signaling networks to regulate plant immunity responses [40]. Our results indicate that the higher the concentration of Alta1, the stronger the immune response triggered, with the content of JA gradually increasing (Figs 1B and S8). This data further determined that the immune response induced by Alta1 includes JA signaling.

This seemingly paradoxical function is not untraceable. In the past few decades, as research on plant pathogens has progressed, several effectors have been thoroughly studied. As key virulence determinants, these effectors are capable of causing host plant tissue death and facilitating host infection, and also triggering plant immune responses. For instance, a GH12 protein, XEG1, secreted by *P. sojae* induces cell death, triggers plant innate immune responses, and degrades the plant cell wall through its enzymatic activity to promote pathogen colonization [41]; *C. shiraiana* secretes the apoplastic effector Cs02526, which induced strong immune responses in plants but also improved pathogenicity, facilitating *C. shiraiana* infection [42]. As effectors, necrosis and ethylene-inducing peptide 1-like proteins (NLPs) act as toxin-like

virulence factors that induce tissue necrosis and trigger plant immune responses [43,44]. These reports indicate the different functions of effectors in pathogenic mechanisms and the induction of immune responses may be through distinct signaling pathways. Our study found Alta1 exhibits dual functions in plant–pathogen interactions. On the one hand, Alta1 was involved in the pathogenic signaling pathways of *A. alternata*, and on the other hand, it activates innate immune response in plants. Here, we have elucidated the signaling pathway by which Alta1 induces plant disease resistance responses, while it is necessary to reveal the molecular mechanisms underlying its contribution to virulence in the future.

## Role of the clock protein CmWD40 in plant immunity

Effectors regulate plant immunity by interacting with crucial host molecules, including signaling components, enzymes, transcriptional regulators and resistance (*R*) genes [45]. Our study on the *Aa*Alta1 effector showed that its interaction with the chrysanthemum gene *CmWD40* is crucial for enhancing plant immunity (Figs 4 and S16).

WD40 proteins contain WD repeat motifs, which are protein sequences composed of 44–60 amino acid residues, and serve as a scaffold for the interaction between two proteins or between proteins and DNA [46]. In addition, these motifs play important roles in many fundamental biological processes, such as signal transduction, histone modification, DNA damage response, transcription regulation, RNA processing, protein degradation, and apoptosis [47]. Consistent with their essential roles, several WD40 proteins trigger immune responses in humans and mammals [48,49]. In plants, the WD40 protein family play important roles in the transcriptional regulation of many processes, including plant growth and stress resistance after being inducted by hormones and the circadian clock [50–55]. In *Arabidopsis*, the homologous protein LWD1 plays an essential role in the transcriptional regulation of the central circadian clock component CCA1, which is critical for binding to TCP20/TCP22 [31]. Previous studies have shown that CCA1 positively modulates resistance during *M. oryzae* infection [56]. Several *R* genes are likely the direct targets of CCA1 based on the overrepresentation of two CCA1 binding motifs (evening element and CCA1 binding site) in this *R* genes promoter [57]. Similar to the expression pattern of *AtLWD1*, *CmWD40* showed rhythmic basal expression, which peaked at dawn (Fig 5A) and affected the expression of *CCA1* and *TCP* genes (S14 Fig), suggesting *CmWD40* may interact with the *TCP* transcription factor to form a complex that activates the transcription of CCA1 in the early morning. Moreover, we found that the resistance of chrysanthemum to *A. alternata* varied with time and aligned with the expression pattern of *CmWD40*, with the strongest resistance observed after inoculation at dawn (Fig 5A). In addition, *Cm*WD40 is induced by *A. alternata* infection (Fig 5B). Collectively, the *Cm*WD40 protein spatiotemporally fine-tuned rhythmic activities to acquire resistance against daily stresses. The temporal control of defense mediated by the circadian clock is represented by rhythmic changes in the levels of defense-related molecules, reflecting the role of the circadian clock in anticipating likely attacks from pathogens [58–60]. For example, in the absence of pathogens, the levels of some defense-related genes, including *FLS2*, *MKK4/MKK5*, *MPK3/MPK6*, and *WRKY22*, and the production of defense signaling molecules, such as SA, JA, and ROS, oscillate with changing peaks throughout the day [61–63].

Subsequently, based on RNA-seq analysis, we further determined that CmWD40 is involved in activating the circadian clock and JA-mediated defense signaling (Fig 6B and 6C), suggesting that increased resistance to *A. alternata* infection observed under *Cm*WD40 overexpression depends on the circadian clock and JA signaling. Previous studies have shown that the clock gene *LUX* regulates JA signaling, and that LUX expression is also influenced by JA, which suggests *LUX* is one of a key nodes in mediating crosstalk between the circadian clock and defense signaling involving JA [63]. A clock gene TIC has been shown to regulate JA

signaling through its direct interaction with the MYC2 protein [64]. Notably, we found that *CmWD40* positively regulated the expression levels of JA-related genes and induced the accumulation of JA (Fig 6). Numerous studies have shown that JA-induced resistance is mainly effective against necrotrophic pathogen infection [65–68]. In addition, exogenous methyl jasmonate induces *A. alternata* resistance in chrysanthemum [69], demonstrating that JA signaling is a positive regulator of chrysanthemum defenses against *A. alternata*. Accordingly, we speculate that the clock component *Cm*WD40 modulates key defense signaling mediated by JA, leading to disease resistance against pathogens.

As is well-known, MYC2 can activate JA-responsive genes and affect particular aspects of JA-regulated resistance against pathogens and stress responses [70,71]. In this study, the results of the qRT-PCR analysis revealed that *Cm*WD40 activated the expression of the *CmMYC2* gene, which exhibited a similar rhythm expression pattern (Figs 6F and S14), suggesting that the circadian clock affects *Cm*MYC2-mediated regulation of JA signaling. Additionally, the overexpression of the *CmWD40* gene or the pretreatment of *Aa*Alta1 conferred enhanced resistance against *A. alternata* in chrysanthemum plants. When *Cm*MYC2 was silenced in the OX-*Cm*WD40 line, *Aa*Alta1–CmWD40-induced resistance against *A. alternata* was weakened (Fig 7B and 7C), implying that *Cm*MYC2 function is important for increasing plant resistance. These findings indicate that the *Aa*Alta1–*Cm*WD40–*Cm*MYC2 interaction positively regulates chrysanthemum resistance to *A. alternata*, with all three genes probably being in the same genetic pathway.

Apart from several JA derivatives, (+)-strigol exhibited similar accumulation patterns in the OX-*Cm*WD40 lines. In addition, the accumulation of ACC showed significant differences between the OX-*Cm*WD40 and WT lines, with or without *A. alternata* inoculation (S17 Fig), which indicates that JA, in conjunction with other plant hormones, is involved in chrysanthemum resistance to *A. alternata* at the chemical level. ACC can be oxidized by ACC oxidases to produce ethylene (ET) [72]. The production of ET can enhance the resistance of litchi to *Peronophythora litchi* [73]. ET can also regulate plant resistance to necrotrophic pathogens, such as *Botrytis cinerea* [74]. These data illustrate that the JA and other phytohormones can be activated by *Cm*WD40 to trigger plant immune responses, but the triggered mechanisms of other phytohormones need further investigation.

In conclusion, we describe a novel effector *Aa*Alta1 in the necrotrophic pathogen, *A. alternata*, that enhances chrysanthemum immunity by targeting the clock protein CmWD40 to activate the JA signaling pathway (Fig 7D). *Cm*WD40 mediates *Aa*Alta1-induced JA signaling by activating the expression of *Cm*MYC2, which further activates defense signal cascades to mount resistance against *A. alternata*. These findings illustrate that the mechanism underlying the pathogen effector-mediated regulation of host plant immunity is complex, and Alta1, as an efficient plant immunity activator, is a candidate target for controlling plant diseases.

## Materials and methods

### Plants and *A. alternata* culture conditions, inoculation

The host plant species *Chrysanthemum morifolium* cv. 'Jinba' and *N. benthamiana* were obtained from the Chrysanthemum Germplasm Resource Preserving Centre of Nanjing Agricultural University, China. The pathogen test strain *A. alternata* was isolated and identified from the typical diseased leaves of 'Fubaiju', a cultivar found in the chrysanthemum tea-producing area of Futianhe Town, Macheng City, Hubei Province, China, in 2017. Plants were grown under controlled conditions (temperature: 28 °C; D/N light duration: 16 h/8 h; light intensity: 200 $\mu$E·m$^{-2}$·s$^{-1}$ during the day; and relative humidity: 70%) in a plant growth chamber. *Alternaria alternata* was cultured on potato dextrose agar medium for 7–15 days at

28 °C, and the fungal cakes were crushed and transferred into 200 mL potato dextrose water liquid medium and grown overnight at 28 °C with shaking at 180 rpm before inoculation assays were performed.

A 1 mL homogeneous mycelium suspension was collected (with the amount of mycelium contained in each milliliter of suspension constant). A fine-bristle brush was used to pick out the mycelium and inoculate chrysanthemum leaves. Each leaf was inoculated at two inoculation sites. Subsequently, it was cultivated in a controlled environment under a photoperiod of 16 h light/8 h dark at 28 °C and 70% humidity. The lesion area was evaluated under the 16 h light/8 h dark photoperiod at 48 hpi.

## Bioinformatics analyses

The transcriptomics analysis of *A. alternata* was used for the preliminary screening of *A. alternata* effectors [28]. The SignalP 5.0 server (https://services.healthtech.dtu.dk/services/SignalP-5.0/) was accessed to predict the presence of SPs and the location of their cleavage sites in proteins. The TMHMM 2.0 server (https://services.healthtech.dtu.dk/services/TMHMM-2.0/) was accessed to predict transmembrane helices in proteins. The big-PI predictor server (https://mendel.imp.ac.at/gpi/gpi_server) was accessed to predict the glycosylphosphatidylinositol-modification sites in proteins. The conserved protein domain search was performed by Pfam 35.0 (http://pfam-legacy.xfam.org/), whereas the databases NCBI and Uni-Prot were used for BLASTp searches. The mature protein sequences were aligned using the ClustalW2 program, and a phylogenetic tree was constructed using MEGA X with a maximum likelihood model.

## Construction of plasmids

The coding sequences of candidate effectors and chrysanthemum genes were amplified from the cDNA of *A. alternata* mycelium and the leaves of the 'Jinba' cultivar, respectively. The purified fragments were cloned into vectors using the ClonExpress II One Step Cloning Kit (Vazyme Biotech Co. Ltd., Nanjing, China) and transformed into the *E. coli* strain DH5α. Each construct was verified by sequencing.

## *Agrobacterium tumefaciens*-mediated transient expression

The pICH31160-effector constructs were introduced into the *A. tumefaciens* strain GV3101 (pJIC SA_Rep) used for transient expression in *N. benthamiana*. *Agrobacterium tumefaciens* carrying various vectors were grown overnight in Luria–Bertani medium supplemented with kanamycin (50 mg·L$^{-1}$) and rifampin (50 mg·L$^{-1}$) at 28 °C/180 rpm. The cultured *A. tumefaciens* cells were collected after centrifugation for 10 min at 28 °C/ 4,000 rpm, washed thrice using an infiltration buffer (10 mM MES, pH 5.7; 10 mM MgCl$_2$; and 20 nM acetosyringone) and incubated in the infiltration buffer for 3 h before infiltration. The suspensions of *A. tumefaciens* cells were infiltrated into 4–6-week-old *N. benthamiana* plants at appropriate sites. Two days after infiltration, leaves were harvested for detecting protein accumulation. Cell death was observed 7 days after infiltration.

## Leaf infiltration assay with purified proteins

The *E. coli* strain BL21 (DE3) was used for protein expression. pET32a-6xHis plasmids harboring *Aa*Alta1 coding sequences (*Aa*Alta1-6xHis) were transformed into BL21 (DE3). Subsequently, the transformed strain was cultured until an optical density of 0.6–0.8 was achieved at 37 °C/180 rpm. Next, isopropyl ß-D-1-thiogalactopyranoside was added until the final concentration was 0.5 mM, and the mix was cultured for 4 h, centrifuged, and the precipitated protein sample was collected. The protein pellet was resuspended in phosphate buffer saline

and centrifuged again. The residue precipitate was dissolved, resuspended in a denaturing buffer, and centrifuged again. The supernatant was purified by Ni column affinity chromatography. To test the cell death-inducing activity of the recombinant proteins, 30 nM-3 μM protein solutions were infiltrated into *N. benthamiana* and chrysanthemum leaves. Cell death was observed 3 days after infiltration.

## Extraction of plant proteins

The plant leaves expressing the recombinant proteins were ground to powder in liquid nitrogen. Subsequently, they were added to 1 mL extraction buffer (10% glycerol; 25 mM Tris pH 7.5; 1 mM EDTA; 150 mM NaCl) to which 10 mM dithiothreitol, 1% protease inhibitor cocktail, and 0.15% NP40 were added. The resulting mix was vortexed and blended uniformly. The samples were centrifuged at 12,000 × *g* for 20 min at 4 °C, and the supernatant was used for SDS-PAGE gel electrophoresis.

## Oxidative burst assay

For DAB staining, the treated leaves were immersed in a DAB solution (0.1 g DAB was dissolved in 100 mL 10 mM pH 7.8 phosphate buffer, and the pH was set to 3.8) and left to stand in the dark at 26 °C for 6-8 h. After the staining, the leaves were soaked in anhydrous ethanol. Subsequently, the residual staining solution was removed, and the leaves were placed in a 10% DAB–glycerol solution, following which the leaves were soaked in ethanol for 4 h. Finally, the images were captured and observed.

For the luminol/ peroxidase assay, leaf discs (the diameter is 0.5 cm) were collected from 5-week-old *N. benthamiana* and then soaked in 200 μL sterile water in a 96-well plate overnight. The sterile water was replaced with 200 μL reaction buffer containing luminol/peroxidase (35.4 mg·mL$^{-1}$ luminol and 10 mg·mL$^{-1}$ peroxidase) and different PAMPs (200 nM flg22 and 1 mM *Aa*Alta1). Luminescence was measured using the GLOMAX96 microplate luminometer (Promega, Madison, WI, USA).

## PEG-mediated fungal genome editing and complementation in *A. alternata*

To generate the AaAlta1 KO construct (*Δalta1*), a 965-bp fragment the of 5′ flanking region and 811-bp of the 3′ flanking region of the *AaAlta1* coding sequence were PCR amplified from *A. alternata* genomic DNA using gene-specific primer pairs. Moreover, the hygromycin phosphotransferase gene cassette was PCR amplified from the pcDNA1 vector. Three fragments were fused into one fragment using fusion PCR and then transformed into the protoplasts of the WT *A. alternata* strain. For the complementation tests of the *Δalta1* mutant (*Δalta1/Alta1*) strain, the native AaAlta1 and neomycin genes were PCR amplified using primers. The amplified fragments were fused with the 5′ and 3′ flanking regions (5′AaAlta1::AaAlta1::NEO::3′AaAlta1), and the complement fragment was transformed into the protoplasts of the *Δalta1* mutant strain. The PEG-mediated protoplast transformation of *A. alternata* was performed following the published protocols [75].

## DNA and RNA isolation and gene expression analysis using qRT-PCR

The fungal DNA kit (OMEGA, USA) and the rapid plant genomic DNA Isolation kit (Waryong, Beijing, China) were used to extract the genomic DNA of *A. alternata* and chrysanthemum, respectively. The plant RNA Isolation kit (Waryong, Beijing, China) was used to extract the RNA of chrysanthemum plants. The isolated RNA was subjected to reverse transcription to obtain cDNA samples using the cDNA synthesis kit (TaKaRa Biomedical Technologies Co. Ltd.), and qRT-PCR assays were performed as previously described [76]. Transcript

abundance estimates were based on the mean of three biological replicates and were calculated using the $2^{-\Delta\Delta Ct}$ method [77]. *Cm*EF1a (GenBank: KF305681.1) and *Aa*Actin (GenBank: XM_018525258.1) were used as internal control reference genes for chrysanthemum and *A. alternata*, respectively. The primers are listed in S1 Table.

## Subcellular localization

The coding sequence of AaAlta1ΔSP and CmWD40 were cloned into the pORE-R4-GFP vector. The *Agrobacterium* strain GV3101 was injected into *N. benthamiana* leaves to transform the combinations of p35S::gene-GFP and 35S::D53-RFP constructs (nuclear markers; RFP, red fluorescent protein). The control samples were transformed with an empty pORE-R4-GFP vector. Infiltrated plants were transferred to a growth chamber at 28 °C under a dark photoperiod for 24 h, and then moved to a 16-h light/8-h dark photoperiod. After 48 h, the fluorescence signal was detected.

## Transient expression of proteins in yeast strain

For the yeast secretion assay, the AaAlta1 SP (the signal peptide of AaAlta1) and PR1 protein (the SP from pathogenesis-related protein 1, used as a positive control) were inserted into the plasmid pSUC2-MSP and transformed into the yeast strain YTK12, which was incubated on CMD/−W (Trp dropout medium containing 0.1% glucose and 2% sucrose) medium and YPRAA (with raffinose as the sole carbon source) medium for 3 days at 30 °C. The reduction of TTC to insoluble red-colored 1, 3, 5-triphenyl formazan enabled the detection of the invertase enzymatic activity. Transformants were cultured in liquid CMD/−W medium, and approximately 1.5 mL of cell suspension was collected and centrifuged to remove the supernatant. Subsequently, the pellets were then resuspended in 250–750 μL sterile distilled water, 10 mM acetic acid–sodium acetate buffer (pH 4.7), and 500 μL 10% sucrose solution (w/v) and incubated at 37 °C for 10 min. Next, the mix was centrifuged at 12,000 × g for 1 min. Approximately 400 μL of the supernatant was added into a fresh 5 mL tube to which 3.6 mL of 0.1% TTC solution was added, and the mix was left to stand at 28 °C for 5 min. The images were captured immediately following the color change.

For the yeast two-hybrid assay, the bait pGBKT7-AaAlta1ΔSP construct was transformed into the yeast strain Y2H to screen the prey chrysanthemum yeast cDNA library following the method described earlier [78]. The coding regions of *Cm*WD40 and *Aa*Alta1ΔSP were cloned into the pGBKT7 and pGADT7 vectors, respectively. Subsequently, the vectors were co-transformed into the yeast strain Y2H, which was incubated on SD/−Trp/−Leu medium for 3 days at 30 °C. The transformed yeast cells were transferred to SD/−His/−Leu/−Trp/−Ade- medium and SD/−His/−Leu/−Trp/−Ade- supplemented with X-α-Gal. The pGBKT7-53 and pGADT7-T vectors were used as positive controls, and the pGBKT7-Lam and pGADT7-T vectors were used as negative controls.

## Bimolecular fluorescence complementation assay

The coding region of AaAlta1 was inserted into the pSPYNE(173) vector, and that of *Cm*WD40 was inserted into the pSPYCE(M) vector [79]. The resulting plasmids were introduced into the *Agrobacterium* strain GV3101, co-infiltrated with the 35S::D53-RFP constructs (nuclear markers) into the leaves of 4-week-old *N. benthamiana* plants. The yellow fluorescent protein signal was observed 3 days after infiltration using a confocal microscope (LSM800; Zeiss, Germany) to observe the yellow fluorescent protein signal following a previously described method [80].

## Pull-down assay

The coding region of the *CmWD40* gene was inserted into the pGEX-5X-1-GST vector, and the plasmids were introduced into the *E. coli* strain BL21(DE3). The *Cm*WD40-GST and AaAlta1ΔSP-6xHis constructs were induced for protein expression. The target protein was obtained via ultrasonic crushing. The *Cm*WD40-GST and *Aa*Alta1ΔSP-6xHis proteins were mixed, and the pGEX-5X-1-GST empty vector was mixed with *Aa*Alta1ΔSP-6xHis as control and incubated at 4 °C for 2 h. Approximately 100 μL of each mix was used as the input. The remaining cells were incubated with GST beads at 4 °C for 4 h and washed thrice with the wash buffer (4.2 mM $Na_2HPO_4$, 2 mM $KH_2PO_4$, 140 mM NaCl, 10 mM KCl). Finally, the eluent from the beads was used as the output. Immunoblot analysis was performed to detect proteins using an anti-His antibody (Thermo, USA).

## Transformation of chrysanthemum

The pORE-R4-flag-CmWD40 plasmid was transformed into the *Agrobacterium* strain EHA105 for the transformation of the chrysanthemum cultivar 'Jinba' using the *Agrobacterium*-mediated leaf disc transformation method, as previously described [81]. Subsequently, the primer pair 35S-F/*Cm*WD40-R was used to identify the transgenic lines at the DNA level using qRT-PCR, and the *Cm*WD40-F/R primers were used to determine the relative expression levels. The generated transgenic plants were transferred from the MS medium to the soil and cultivated in a field. Next, the cuttings were collected and cultivated, and prepared for the subsequent experiments.

## Measurement of JA content

The leaves of the chrysanthemum cultivar 'Jinba' were collected from different lines at the same growth stage and quickly ground into powder using liquid nitrogen. Subsequently, 0.02 g of samples were placed into a 2 mL centrifuge tube and then homogenized in 1 mL extracting solution (methanol: acetic acid=99:1) using vortexing. The samples were then centrifuged for 5 min at 13,000 rpm and 4 °C, and the supernatant was collected and passed through the 0.45 μm filter membrane. High-performance liquid chromatography coupled with tandem mass spectrometry was used to measure JA content, and the internal standard settings were set as described previously [82].

## Measurement of CD spectrum

Prepare 300 μL of a protein solution with a 0.5 μg/μL concentration. Transfer to a 10 kDa ultrafiltration tube and centrifuge at 4 °C with 13,000 rpm for 10 min. Add 200 μL $ddH_2O$, and centrifuge at 4 °C with rpm for 10 min. Repeat this step 3 times and collect the fraction>10kD. Then, dilute the sample with $ddH_2O$ to a of 0.1 μg/μL concentration. Use a circular dichroism spectrometer (BTP-E130) for far UV detection of CD spectrum.

## Supporting information

**S1 Fig. Screening process of candidate effectors in *Alternaria alternata*.** (A) High-throughput screening of candidate effectors using transcriptomics and bioinformatics analyses. (B) Transient transformation of candidate effectors in *Nicotiana benthamiana* leaves. *Nicotiana benthamiana* leaves were infiltrated with *Agrobacterium tumefaciens* strain GV3101 cells harboring the *A. alternata* effector genes, the *Phytophthora infestans* gene *INF1* and the empty vector (EV). The *Agrobacterium* cells harboring the *INF1* gene and EV were used as positive and negative controls, respectively. Partial results have been presented.
(TIF)

**S2 Fig. *Aa*Alta1 induced multiple immune responses.** (A) Sequence alignment of the protein sequence of the *Aa*Alta1 protein with that of its ortholog effector from *Verticillium dahliae* (PevD1, VDAG_02735). (B) Relative expression of immune marker genes triggered by *Aa*Alta1 in *Nicotiana benthamiana* leaves. Expression levels were assessed by quantitative reverse transcriptase polymerase chain reaction. Data are presented as mean ± standard error of three biological replicates. Different letters at the top of error bars indicate significant differences ($P$ <0.05, Tukey's test). (C) *Aa*Alta1 induces reactive oxygen species burst in *N. benthamiana*. DAB staining and decolorization were performed 3 days after the *Agrobacterium* treatment. The *Agrobacterium* strain harboring the *INF1* gene was used as a positive control, and EV was used as a negative control in *N. benthamiana*.
(TIF)

**S3 Fig. Multiple sequence alignment of AaAlta1 with 11 orthologs from other *Alternaria alternata* strains.** *Alternaria arborescens* (XP_028503382.1), *Alternaria gaisen* (KAB2107613.1), *Alternaria burnsii* (XP_038782547.1), *Alternaria panax* (KAG9189691.1), *Alternaria atra* (XP_043172693.1), *Alternaria dauci* (XP_069306948.1), *Alternaria solani* (WPZ50800.1), *Alternaria hordeiaustralica* (XP_049246647.10), *Alternaria metachromatica* (XP_049193700.1), *Alternaria incomplexa* (XP_051290017.1), *Alternaria irosae* (XP_046029020.1).
(TIF)

**S4 Fig. Comparison of the experimental spectrum (black dashed line) and the predicted spectra produced using the (green line) KCD servers for the Alta1 protein.**
(TIF)

**S5 Fig. Generation and identification of AaAlta1 knockout (KO) mutant and complementation strains.** (A) Schematic diagram of the homologous recombination-based AaAlta1 KO (*Δalta1*) and complementation (*Δalta1/Alta1*) strategy. In the KO construct, the hygromycin phosphotransferase gene is flanked by 966 bp of the 5′ sequence and 812 bp of the 3′ sequence of the gene region of *AaAlta1*. In the complementation construct, *AaAlta1* and the neomycin resistance gene were flanked by the 5′ sequence and 3′ sequence of the *AaAlta1* gene region. (B) Polymerase chain reaction-based verification of the construction of replacement cassette. (C, D) WT, *Δalta1,* and *Δalta1/Alta1* strains were identified at the DNA level.
(TIF)

**S6 Fig. The reduced virulence of Δalta1 mutants is not primarily due to a decrease in growth.** (A) Symptoms of the disease on chrysanthemum leaves inoculated with WT at mycelial normal amounts, and two *Δalta1* (*Aa*Alta1 knockout mutant) strains at mycelial double amounts. Images were captured at 48 hpi. + represents inoculated with 1 mL collected homogeneous mycelium suspension. Scale bar = 1 cm. (B) Disease severity was determined by measuring the lesion area (mm$^2$) of leaves 48 hpi. Data are presented as the mean ± standard error of three biological replicates. *** $P$ ≤0.0001 compared with control, as calculated by one-way analysis of variance.
(TIF)

**S7 Fig. Alta1 is important for the full virulence expression of *Alternaria alternata* (tobacco pathotype).** (A) Symptoms of the tobacco brown spot disease on *Nicotiana benthamiana* leaves inoculated with WT and *Δalta1* (*Aa*Alta1 knockout mutant) strains. Images were captured at 24 hpi. Scale bar = 1 cm. (B) Disease severity was determined by measuring the lesion area (mm$^2$) of leaves 24 hpi. Data are presented as the mean ± standard error of three biological replicates. **** $P$ ≤0.00001 compared with control, as calculated by one-way analysis of variance.
(TIF)

**S8 Fig. Measurements of jasmonic acid (JA) content in chrysanthemum leaves infiltrated with Alta1 protein.** Representative chrysanthemum leaves infiltrated with purified Alta1 protein (150 nM to 1 mM) or mock solution. Measurements of JA content after 24 h. Data are presented as the mean ± standard error of three biological replicates. Different letters at the top of error bars indicate significant differences (*P* <0.05, Tukey's test).
(TIF)

**S9 Fig. Sequence alignments and phylogenetic analysis of *Cm*WD40.** (A) Polymerase chain reaction product of the cloned *CmWD40* gene. (B) Multiple sequence alignment of the CmWD40 and homologous WD40 proteins of other species, i.e., *Artemisia annua* (AaWD40, ATX64403.1), *Erigeron canadensis* (EcWD40, XP_043631766.1), *Helianthus annuus* (HaWD40, XP_021973454.1). The WD40 repeat domain is marked with red lines. (C) Phylogenetic tree of the CmWD40 and other WD40 proteins, including *Artemisia annua*, *Helianthus annuus*, *Erigeron canadensis*, *Nicotiana attenuata* (NaLWD1, XM_019389340.1), and *Arabidopsis thaliana* (AtLWD1).
(TIF)

**S10 Fig. (A) Yeast two-hybrid assay demonstrates protein–protein interactions between *Cm*WD40 and different deletion fragments of *Aa*Alta1.** SD-4, SD/-His/-Leu/-Trp/-Ade- medium; SD-2, SD/-Leu/-Trp- medium. (B) Images showing the phenotypes of leaves infiltrated with *Agrobacterium* strains harboring constructs encoding the different deletion fragments of *Aa*Alta1 3 days after infiltration.
(TIF)

**S11 Fig. Yeast-two-hybrid assay showing that *Aa*Alta1 interacts with *Nb*WD40.** SD-4, SD/-His/-Leu/-Trp/-Ade- medium; SD-2, SD/-Leu/-Trp- medium.
(TIF)

**S12 Fig. Identification of the Alta1-GFP (green fluorescent protein) fusion strains.** (A) Identification of Alta1-GFP fusion strains at the DNA level. (B) Relative expression of GFP in different fluorescent strains. (C) Confocal microscopy images showing fluorescent hypha in the Alta1-GFP fusion strain. (D) Subcellular localization of the fusion protein in the Alta1-GFP fusion strain.
(TIF)

**S13 Fig. Subcellular localization of *Cm*WD40 and *Aa*AltA1 in *Nicotiana benthamiana* leaves.** The co-expressed 35S::D53-RFP (red fluorescence protein) construct was used as a nuclear marker. Marker: images taken in the red fluorescence channel; GFP: green fluorescent protein (image captured in the green fluorescence channel); DIC: differential interference contrast (image captured in the bright light channel); merged: both overlay plots. Scale bars = 20 μm.
(TIF)

**S14 Fig. Rhythmic expression patterns of genes in WT and overexpression (OX)-*Cm*WD40 lines.** Chrysanthemum plants were cultured in LD conditions for 4 weeks, and samples were collected every 6 h at the indicated times. Gene expression levels were assessed by quantitative reverse transcriptase polymerase chain reaction. Data are presented as mean ± standard error of three biological replicates.
(TIF)

**S15 Fig. *Cm*WD40 is involved in the transcriptional activation of rhythm genes.** (A) Heat-maps of differentially expressed genes (DEGs) encoding clock-related genes in overexpression (OX)-*Cm*WD40 transgenic chrysanthemum, as assessed using RNA-seq. The gene expression

values are normalized $\log_2$(FPKM [fragments per kilobase of transcript per million fragments mapped] + 1). From left to right are shown WT, OX-CmWD40, WT-I, and OX-CmWD40-I. WT, control, non-infected plants (WT cultivar 'Jinba'); OX-WD40, non-infected OX-*Cm*WD40 transgenic plants; WT-I, control, infected plants; OX-WD40-I, infected OX-*Cm*WD40 transgenic plants. (B) FPKM of CCA1 in RNA-Seq.
(TIF)

**S16 Fig. Heatmaps of differentially expressed genes encoding defense-related proteins in overexpression (OX)-*Cm*WD40 transgenic chrysanthemum, assessed using RNA-seq.** The gene expression values are normalized $\log_2$(FPKM [fragments per kilobase of transcript per million fragments mapped] + 1). From left to right are shown WT-I, OX-*Cm*WD40-I. WT-I, control, infected plants; OX-WD40-I, infected OX-*Cm*WD40 transgenic plants.
(TIF)

**S17 Fig. Heatmaps analysis of metabolites.** The horizontal axis represents the names of samples, whereas the vertical axis represents the information of metabolites. Different colors indicate different content levels, with the colors filled based on the *Z*-score normalized values (red represents high content, and green represents low content). Cluster analysis was performed on the metabolites, with the clustering lines on the left side of the figure representing the metabolite clustering lines.
(TIF)

**S18 Fig. Differentially expressed genes (DEGs) co-upregulated by Alta1 and *Alternaria alternata* infection.** (A) Venn diagram analysis of the DEGs induced by Alta1 and *A. alternata*. (B) Kyoto Encyclopedia of Genes and Genomes enrichment analysis of the DEGs co-upregulated by Alta1 and *A. alternata*.
(TIF)

**S19 Fig. Heatmaps of differentially expressed genes (DEGs) encoding isoflavonoid biosynthesis-related genes in RNA-seq.** The gene expression values are normalized $\log_2$ (FPKM [fragments per kilobase of transcript per million fragments mapped] + 1). From left to right are shown the mock and Alta1. Mock, heat map of the DEGs in mock-infiltrated tissues; Alta1, heat map of the DEGs in 300 nM Alta1-infiltrated tissues.
(TIF)

**S1 Table. Primers used in this study for quantitative reverse transcriptase polymerase chain reaction.**
(XLSX)

## Acknowledgments

We thank Suomeng Dong from Nanjing Agricultural University for providing the binary vector pICH31160. We thank Xiaojie Wang from Northwest A&F University for providing the binary vector pSUC2 and yeast strain YTK12.

## Author contributions

**Conceptualization:** Shuhuan Zhang, Ye Liu, Zhiyong Guan, Jiafu Jiang.

**Data curation:** Shuhuan Zhang, Lina Liu, Wenjie Li, Mengru Yin.

**Formal analysis:** Lina Liu, Wenjie Li, Jiafu Jiang.

**Funding acquisition:** Zhiyong Guan.

**Investigation:** Shuhuan Zhang.

**Methodology:** Shuhuan Zhang, Lina Liu, Qian Hu, Jiafu Jiang.

**Project administration:** Zhiyong Guan, Jiafu Jiang.

**Writing – original draft:** Shuhuan Zhang.

**Writing – review & editing:** Sumei Chen, Fadi Chen, Ye Liu, Zhiyong Guan, Jiafu Jiang.

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
