## [Decision Letter · Decision Letter 0]

8 Aug 2024

Dear Drs. Liu, Guan, and Jiang,

Thank you very much for submitting your manuscript "Alternaria alternata effector AaAlta1 targets CmWD40 and participates in regulating disease resistance in Chrysanthemum morifolium" for consideration at PLOS Pathogens. As with all papers reviewed by the journal, your manuscript was reviewed by members of the editorial board and by several independent reviewers. In light of the reviews (below this email), we would like to invite the resubmission of a significantly-revised version that takes into account the reviewers' comments.

While the judgements of the reviewers ranged significantly in their overall view of the manuscript, the general consensus was that the work conducted was of sufficient quality and interest for the readership of PLOS Pathogens. However, the authors should take careful consideration of the suggestions of all three reviewers. The manuscript itself requires a very careful rewrite considering significant grammatical issue throughout.

We cannot make any decision about publication until we have seen the revised manuscript and your response to the reviewers' comments. Your revised manuscript is also likely to be sent to reviewers for further evaluation.

Sincerely,

Melvin Bolton

Guest Editor

PLOS Pathogens

Bart Thomma

Section Editor

PLOS Pathogens

Michael Malim

Editor-in-Chief

PLOS Pathogens

orcid.org/0000-0002-7699-2064

The manuscript by Zhang and colleagues has been reviewed by three experts in the field. While the judgements of the reviewers ranged significantly in their overall view of the manuscript, the general consensus was that the work conducted was of sufficient quality and interest for the readership of PLOS Pathogens. However, the authors should take careful consideration of the suggestions of all three reviewers. The manuscript itself requires a very careful rewrite considering significant grammatical issue throughout.

Reviewer's Responses to Questions

**Part I - Summary**

Reviewer #1: Strengths: Well thought out experiments, full circle analysis of host and pathogen interaction, omics and functional genetic based approach, data presented are clear and graphs are well constructed. Its very nice work.

Weaknesses: Use of only 1 strain of the fungus, 1 mutant strain and no ectopic or non-complemented control strains were included for analysis. No chemical analysis to nail down phytohormones in the host.

Novelty/significance: Low: effector biology has been beaten to death and become pedestrian in the arena of host-parasite interactions. While the findings may be new for the alternaria-cut flower context, they are not so impactful in the larger context of plant-fungal interactions.

General execution and scholarship: good, but i would re-do some of the figures that appear to have been cut and pasted together. Re-take all as one photo. While i am in NO WAY implicating scientific chicanery, I am suggesting that it makes the findings for that particular experiment in doubt for the most critical reviewer/readership.

Reviewer #2: Zhang et al. present a comprehensive functional characterization of the effector AaAlta1 and dissect its effect on the chrysanthemum host immune response. Interestingly, the Alta1 protein from Alternaria alternata was first identified as being an allergen in human respiratory disease, and it appears to play a similar role in plant systems. They demonstrate through functional studies of knockout mutants, transcriptomics and protein-protein interaction assays that Alta1 induces the expression of and interacts with CmWD40 to stimulate a downstream signalling cascade where the transcription factor MYC2 is activated and jasmonic acid signalling occurs. Ultimately, Alta1 induces programmed cell death via the activation of immune pathways. However, the absence of Alta1 reduces disease symptoms and this should be discussed further- what is the role of Alta1 in the Alternaria-chrysanthemum disease outcome? Is it acting as a necrotrophic effector and triggering susceptibility? This is important for breeding in chrysanthemum and should be discussed in such a context. Experiments were expertly performed to elucidate the mechanism of action of AaAlta1 but the manuscript requires some rewording and refocusing before publishing.

Reviewer #3: In this manuscript, the authors came to the main conclusion that the A. alternata effector Alta1 contributes to fungal virulence in chrysanthemum by targeting the host WD40-repeat (WDR) protein CmWD40. This acts as regulator of plant immunity in a circadian clock-depending manner by triggering MYC2 transcriptional activation which regulates accumulation and signaling transduction of jasmonic acid-related defense responses.

This conclusion is strongly supported by evidence of protein-protein interaction and co-localization in the host cell nucleus, expression analysis of plant defense related markers and disease assays with fungal transformants and transgenic plants.

**Part II – Major Issues: Key Experiments Required for Acceptance**

Reviewer #1: 1. Do a genome scan for this locus in other A. alternata strains - see how prevalent it is within the species and do MSA to show homology. How well conserved is this gene and encoded protein?

2. Do functional work in at least 1 additional strain of the fungus - perhaps a type specimen or NRRL strain

3. Generate additional mutant strains for current and additional strain, and include complemented, ectopic and non-complemented controls strains in analysis

4. Consider altering the model diagram to include other aspects of fungal infection and plant defense as it appears way too simplistic based on the conclusions of the current investigation.

5. The paper would also benefit from using a targeted metabolomics approach to verify JA and other phytohormone involvement at the chemical level.

Reviewer #2: (No Response)

Reviewer #3: The first and major remark is that the manuscript would benefit from an extensive grammatical cleanup and from a more accurate use of references.

Secondly, a second effector deletion mutant should be tested. The authors used two complemented isolates but only one deletion mutant.

Thirdly, the question how the interaction of Alta1 effector with WD40 can trigger cell death was not addressed. Is this interaction required to trigger cell death or does it occur via a different mechanism? Could a certain JA gradient might be responsible for the observed cell death response upon effector infiltration? Was the protein conformation investigated with crystallography/CD spectroscopy to ensure that the purified protein used for infiltration assay does match the in silico predicted effector protein structure?

Finally, to confirm that during the real host-pathogen interaction, the effector is delivered by the fungus to the assumed (sub-)cellular compartment, an additional experiment is required. Would it be possible to perform an in vivo inoculation assay with a fungus expressing the effector fused to a reporter (e.g. fluorescence marker such as GFP) to check effector delivery and sub-cellular localization in N. benthamiana and/or Chrysanthemum?

**Part III – Minor Issues: Editorial and Data Presentation Modifications**

Reviewer #1: I did not find anything glaring.

Reviewer #2: Minor:

- Change A. alternate to A. alternata throughout

- Correct grammatical errors and typos throughout – but can be done in editing stage.

Introduction:

-Please refocus the introduction to talk about the discovery of the allergen protein Alta1 and previous studies performed in A. alternata (and homologs in other fungal species) instead of paragraphs about WD40 and circadian regulation of host immunity – discussion of the host interactors can be in the discussion. You could move some of the discussion text on Alta1 to the intro. Much work has been done to identify and characterize the role of Alta1 in respiratory inflammatory diseases but this is one of the first comprehensive studies to characterize its role in plant pathogenesis – this should be emphasized. Garrido-Arandia et al. 2016 (https://www.nature.com/articles/srep33468) did show its capacity to bind flavonoids and proposed a plant virulence mechanism based on its structure, and Gomez-Casado et al. 2014 (https://www.sciencedirect.com/science/article/pii/S0014579314001641) demonstrated its ability to bind to and inhibit PR5! Much work done on homologs in plant pathogens e.g. characterization of the Verticillium Alta1 homolog PevD1.

-Could introduce the concept of “effector-triggered susceptibility”

-I would suggest removing any mention of the host targets and moving it to the discussion.

Lines 52-3 “pathogen-associated molecular patterns”

Line 59 – re-phrase this- maybe remove “at this stage” and change to plants “have evolved” resistance (R) genes.

Line 68 – include a reference for “leading to severe yield losses in chrysanthemum production every year”

Line 109- you cannot claim that Alta1 is indispensable for virulence, since disease symptoms still occurred:

“The Δalta1 mutant strain produced mild disease symptoms on chrysanthemum leaves compared to the wild-type strain (Fig 2B), with a lesion area approximately reduced by 70% (Fig 2C).”

Methods:

- Methods (using data from prev paper) – candidate effector identification from RNAseq data. I was surprised that authors didn’t use EffectorP to predict – probably missed many “unique” effectors for Alternaria that are small secreted proteins, since methodology relied upon comparison of gene and domain sequences within existing databases. Were they looking for conserved/”core” effectors? If so, this should be clarified! This is not the current study, so cannot change methods of course. A screen for effectors inducing programmed cell death was then performed through transient expression in Nicotiana – ultimately they are trying to identify conserved effectors that trigger susceptibility?

Results:

- Alta1 appears to be important for normal fungal physiology since the knockout has reduced fitness in the form of slower radial plate growth – role in “normal” fungal physiology could be discussed below

- A key result is that the absence of Alta1 leads to reduced disease symptoms and this is not discussed further – is the triggering of programmed cell death promoting the proliferation of Alternaria? Or is the fitness penalty caused by loss of Alta1 leading to reduced disease?

- Wonderful work elucidating the mechanism of triggering PCD – I am not an expert in these assays but I know they are extremely time-consuming, and they have been all been performed with appropriate controls, replicates and statistical methods of assessments.

- Is there a homolog of this WD40 protein in humans/mammals too (triggering immune response)?

Discussion:

- Alta1 is an example of necrotrophic effector-triggered susceptibility?

- How should this knowledge be used to breed for disease resistance/tolerance in chrysanthemum?

Line 471-2 – “which also suggested that different expressions indicated of AaAlta1 may influence the virulence of A. alternata” – perhaps remove or re-word this.

Line 473-4 – “suggesting that AaAlta1 has complicated pathogenic roles in infection” – perhaps expand on this…

Line 501 – “serval” – wrong word?

Data availability:

 Provide raw data from all experiments in a public archive like figshare.com E.g. raw gel images, yeast two-hybrid data, bimolecular fluorescence complementation data, pull-down assay data, qPCR expression data etc.

 Sequencing data - BioProject IDs cannot be found – not publicly available yet? Make sure that they are made available prior to publishing.

Reviewer #3: Line 133: “Purified Alta1 protein was tested for cell death activity by infiltrating 300 nM to 3 μM protein solution into the mesophyll of N. benthamiana leaves”. How can the authors be sure that the protein solution was infiltrated exactly into the mesophyll layer?

Line 144: “The ability of expression Alta1ΔSP to induce cell death was significantly reduced in N. benthamiana. In contrast, expression of Alta1 did trigger cell death (Fig 1C), suggesting that the ability of AaAlta1 to induce cell death are relyies on the SP”

I don’t fully agree with that statement. The SP is not required to cell death induction but only to protein secretion. However, when looking at Fig. 1C, necrotic tissue is visible at the infiltration spot. Can you please elaborate on that?

In Fig. 1A it is not clear whether the mature effector does consist only in the Allergene motif highlighted in green or there are other motifs which required (or not) for effector activity in terms of interaction with CmWD40 and cell death induction?

In Fig. 1 description, line 185. I guess the authors meant that the SP of PR1 protein served as positive control

Line 207 mentions that “These results demonstrate that Alta1 is an essential virulence factor during the A. alternata infection stage”. I wonder which stage do you mean here.

Line 227: from your transcriptome analysis it is not clear whether the authors did check DEGs just from infiltrated tissue or from Alta1 pre-infiltrated tissue subsequently inoculated with the fungus.

Moreover, would it be possible to have some information on DEGs host genes in inoculated vs non-inoculated tissue?

In Fig. 3A, I don’t entirely understand which is the difference between Mock and Control. Moreover Fig. 3D shows increased transcript levels of isoflavonoid biosynthesis related genes. These might represent important phytoalexins. Do the authors think that it might be worthy to elaborate on that?

Line 276: the authors previously mentioned that the SP is required for effector activity. I wonder how the effector can now localize at the nucleus and interact with CmWD40 if expressed without SP. Could the author elaborate on that? how was the localization experiment conducted?

Fig. 4. Overall, the figure panel do not match with the legend. Can this please be adjusted?

In Fig. 5, I wonder if it would be wise to combine the graph in Fig S6 with the two graphs shown in Fig. 5A and B. Would it make sense to plot both transcripts level of CmWD40 in mock plants and in infected plants either as line or as histogram (Fig 5B)?

Fig. 5E D and H should indicate the time point at which transcripts level were measured.

Line 484: Since Alta1 also caused necrosis in N. benthamiana, it would be wise to test the interaction between Alta1 and NbWD40?

In Material and Methods, I would suggest including information about the A. alternata isolate used. Isolate name/code, where it was collected and so on.

Line 594: How do you mean “plants were inoculated with a quantitative amount of mycelium”?

Line 701: the “previously described chrysanthemum transformation method” requires a reference.

As a final curiosity I’d like to ask the following question:

How conserved is Alta1 effector within the genus Alternaria? Can you speculate on its role in other pathosystems?

PLOS authors have the option to publish the peer review history of their article (what does this mean? ). If published, this will include your full peer review and any attached files.

**Do you want your identity to be public for this peer review?** For information about this choice, including consent withdrawal, please see our Privacy Policy .

Reviewer #1: No

Reviewer #2: No

Reviewer #3: **Yes: ** Michele C. Malvestiti
---

## [Decision Letter · Decision Letter 1]

16 Jan 2025

PPATHOGENS-D-24-01027R1

Alternaria alternata effector AaAlta1 targets CmWD40 and participates in regulating disease resistance in Chrysanthemum morifolium

PLOS Pathogens

Dear Dr. Jiang,

Thank you for submitting your manuscript to PLOS Pathogens. After careful consideration, we feel that it has merit but does not fully meet PLOS Pathogens's publication criteria as it currently stands. Therefore, we invite you to submit a revised version of the manuscript that addresses the points raised during the review process.

Please submit your revised manuscript within 30 days Mar 17 2025 11:59PM. If you will need more time than this to complete your revisions, please reply to this message or contact the journal office at plospathogens@plos.org. Please include the following items when submitting your revised manuscript:

We look forward to receiving your revised manuscript.

Kind regards,

Melvin Bolton

Academic Editor

PLOS Pathogens

Bart Thomma

Section Editor

PLOS Pathogens

Sumita Bhaduri-McIntosh

Editor-in-Chief

PLOS Pathogens

orcid.org/0000-0003-2946-9497

Michael Malim

Editor-in-Chief

PLOS Pathogens

orcid.org/0000-0002-7699-2064

**Reviewers' Comments:**

Reviewer's Responses to Questions

**Part I - Summary**

Reviewer #3: Dear Authors,

after reviewing the second version of the manuscript, I appreciate the effort you put to address my concerns by conducting additional experiments to strengthen the validity of your results. Most of them have been clearly elucidated and added very interesting aspects to the research you conducted.

However, I have to point out that the section Discussion appears to me as a summary of the findings listed in the section results. I don't see any attempt to elaborate on the biological meaning of your scientific discoveries and to connect the information you collected. The manuscript shows a considerable amount of interesting data but, in my opinion, there is no biological story line that can shed light on the mode of action of the effector protein in a contest of plant immunity. Also, I don't see any proposed follow-up research approach to investigate the biological function of the Alta1 effector in the manipulation of JA-dependent host defense responses.

For example, as follow-up research question it would be interesting to investigate how the JA related defense responses are affected by the interaction between Alta1 and the clock regulatory element CmWD40.

**Part II – Major Issues: Key Experiments Required for Acceptance**

Reviewer #3: If the A. alternata Alta1 K.O. lines exhibit reduced radial growth on plates, do you find reasonable to assume that the reduction in lesion size observed on inoculated plants is largely related to the contribution to fungal virulence of the investigated effector?

Maybe I misunderstood that but you often stated that your findings indicate that the AaAlta1 – CmWD40 – CmMYC2 interaction positively regulates chrysanthemum resistance to A. alternata. But then why is the fungus less virulent when you knock out Alta1? What does this contribution rely on?

Based on your comments in the rebuttal letter, unfortunately I still not entirely understand how the SP of the effector protein could be involved in triggering host cell death.

Line 187 "In contrast, the expression of Alta1 did trigger cell death (Fig 1C), suggesting that Alta1 may

possess specific regions that may help to enter into plant cells." this statement does not seem to be related to cell death inducing capability of Alta1. Moreover, how does it trigger cell death if the plant exhibit enhanced resistance upon Alta1 overexpression? Is a certain JA gradient involved in triggering host cell death?

**Part III – Minor Issues: Editorial and Data Presentation Modifications**

Reviewer #3: (No Response)

PLOS authors have the option to publish the peer review history of their article (what does this mean? ). If published, this will include your full peer review and any attached files.

**Do you want your identity to be public for this peer review?** For information about this choice, including consent withdrawal, please see our Privacy Policy .

Reviewer #3: No

**Figure resubmission:**
---

## [Editor Report · Decision Letter 2]

28 Jan 2025

Dear Dr. Jiang,

We are pleased to inform you that your manuscript 'Alternaria alternata effector AaAlta1 targets CmWD40 and participates in regulating disease resistance in Chrysanthemum morifolium' has been provisionally accepted for publication in PLOS Pathogens.

Best regards,

Melvin Bolton

Academic Editor

PLOS Pathogens

Bart Thomma

Section Editor

PLOS Pathogens

Sumita Bhaduri-McIntosh

Editor-in-Chief

PLOS Pathogens

orcid.org/0000-0003-2946-9497

Michael Malim

Editor-in-Chief

PLOS Pathogens

orcid.org/0000-0002-7699-2064
---

## [Editor Report · Acceptance letter]

Dear Dr. Jiang,

We are delighted to inform you that your manuscript, "Alternaria alternata effector AaAlta1 targets CmWD40 and participates in regulating disease resistance in Chrysanthemum morifolium," has been formally accepted for publication in PLOS Pathogens.

Best regards,

Sumita Bhaduri-McIntosh

Editor-in-Chief

PLOS Pathogens

orcid.org/0000-0003-2946-9497

Michael Malim

Editor-in-Chief

PLOS Pathogens

orcid.org/0000-0002-7699-2064